# AGENTSYNTH: SCALABLE TASK GENERATION FOR GENERALIST COMPUTER-USE AGENTS

**Jingxu Xie, Dylan Xu, Xuandong Zhao,**[*] **Dawn Song**
University of California, Berkeley
{jingxuxie,dylanx26,xuandongzhao,dawnsong}@berkeley.edu

## ABSTRACT

We introduce **AgentSynth**, a scalable and cost-efficient pipeline for automatically synthesizing high-quality tasks and trajectory datasets for generalist computer-use agents. Leveraging information asymmetry, AgentSynth constructs subtasks that are simple during generation but significantly more challenging when composed into long-horizon tasks, enabling the creation of over 6,000 diverse and realistic tasks. A key strength of AgentSynth is its ability to precisely modulate task complexity by varying the number of subtasks. Empirical evaluations show that state-of-the-art LLM agents suffer a steep performance drop, from 18% success at difficulty level 1 to just 4% at level 6, highlighting the benchmark's difficulty and discriminative power. Moreover, our pipeline achieves a low average cost of $0.60 per trajectory, orders of magnitude cheaper than human annotations. Our code and data are available at https://github.com/sunblaze-ucb/AgentSynth.

## 1 INTRODUCTION

Large language models (LLMs) have recently shown promise as autonomous agents capable of solving complex, multi-step tasks across a wide range of domains. These LLM agents interact with an environment through structured actions such as mouse clicks, keystrokes, or code executions, and are prompted to complete specific tasks using tools provided by the interface. This paradigm has been explored for web navigation tasks (Yao et al., 2023a; Drouin et al., 2024; Furuta et al., 2023), software development (Yang et al., 2024), formal mathematics (Lin et al., 2025), and many others (Boisvert et al., 2024; Agashe et al., 2025). As research in LLM agents progresses, the availability of high-quality datasets tailored to these domains becomes increasingly critical.

General computer-use tasks that involve interacting with desktop environments and software applications pose especially difficult challenges for data collection. Existing datasets in this space, such as $\tau$-bench (Yao et al., 2024), TheAgentCompany (Xu et al., 2024), OSWorld (Xie et al., 2024), WorkArena (Drouin et al., 2024) rely heavily on human demonstrations over a limited set of tools and tasks. While effective in showcasing agent capabilities, this human-in-the-loop approach is labor-intensive, expensive, and fundamentally unscalable, making it impractical for covering the full breadth of real-world computing scenarios.

To overcome these limitations, recent work has turned to synthetic data generation using LLMs. However, existing pipelines face two core challenges: (1) current LLM agents struggle to generate reliable trajectories for complex tasks, and (2) simplistic or repetitive generation strategies limit task diversity. These challenges are especially acute in visually grounded or long-horizon tasks, where agents must maintain contextual awareness, reason over multiple steps, and adapt when plans fail (Xie et al., 2024; Bonatti et al., 2024). Moreover, limited task diversity increases the risk of overfitting or model collapse during downstream training (Shumailov et al., 2024).

We introduce AgentSynth, a scalable and flexible pipeline for synthesizing diverse, high-quality datasets for training and evaluating computer-use agents. The core insight behind AgentSynth is to exploit information asymmetry between the data generation and evaluation phases, the idea that solving a task step-by-step in the forward direction is far easier than reasoning out the entire solution

---

[*]Corresponding author

all at once. Therefore, we construct the task through a sequence of simple, solvable subtasks. Each subtask builds incrementally on the prior state, with the corresponding trajectories collected during execution. A summarization agent then merges the subtasks into a composite long-horizon task, producing realistic scenarios that are easy to generate but hard to solve.

This design offers several key advantages. By constructing complex tasks from simple, solvable components, AgentSynth enables reliable trajectory collection while maintaining benchmark difficulty. Varying the chaining of subtasks induces combinatorial task diversity. The pipeline is fully automated and achieves a low cost of just $0.60 per trajectory. While we generate over 6,000 tasks in this work, the approach readily scales to tens of thousands of realistic tasks across diverse environments, unlocking significant potential for agent training and evaluation.

Our contributions are as follows:

- We introduce AgentSynth, a fully automated pipeline that synthesizes challenging and diverse computer-use tasks by iteratively chaining LLM-generated subtasks
- We demonstrate how information asymmetry between generation and execution improves trajectory reliability and task complexity, enabling fine-grained task difficulty control.
- We build a benchmark using AgentSynth and show that state-of-the-art agents struggle significantly, revealing a large room for future improvement.

We describe our methodology in detail in Section 3, analyze the generated tasks and datasets in Section 4, and present empirical evaluation results in Section 5.

## 2 RELATED WORK

Substantial research has focused on synthesizing data to improve the training and evaluation of LLMs. However, most existing datasets and benchmarks for computer-use agents still rely heavily on manual design and annotation, limiting their scalability and diversity.

**Synthetic Data Generation.** Synthetic data generation has emerged as a promising approach to enhance model performance and foster new capabilities. Many recent studies have leveraged LLMs to automate and diversify data generation. For instance, Yuan et al. (2025) curated diverse, high-quality datasets extracted from extensive pretraining corpora. Xu et al. (2025a) (Evol-Instruct), Su et al. (2025) (Learn-by-interact), and Sun et al. (2025) (OS-Genesis) both use sequential pipelines to generate synthetic datasets. However, OS-Genesis and Learn-by-interact retroactively define a task over a trajectory instead of stringing together subtasks, while Evol-Instruct only generates the trajectory with the final instruction. Shin et al. (2019) generated synthetic datasets with controlled distributions over programs and specifications. Li et al. (2025) employed an optimization loop where a data generator continuously produces challenging problems targeted at specific evaluation models. Many other applications of synthetic data for LLMs are listed in Liu et al. (2024). These works highlight the power of synthetic pipelines but focus primarily on static text benchmarks rather than interactive agents.

**Agent Datasets and Benchmarks.** Current datasets and benchmarks for agents predominantly depend on human annotators for task creation, demonstration provision, and the definition of evaluation metrics (Yao et al., 2024; Xu et al., 2024; Zhou et al., 2024), which are costly to scale and often limited in diversity. More recent work explores using LLMs to generate agent tasks and trajectories. For example, Pahuja et al. (2025); Trabucco et al. (2025); Murty et al. (2025); Gandhi & Neubig (2025) employed LLMs as web agents to synthesize web-based interactions. Boisvert et al. (2024) composed atomic tasks from Drouin et al. (2024) to form difficult tasks. Xu et al. (2025b) and Ou et al. (2024) turned online tutorials into tasks and demonstrations. Nonetheless, these generated tasks and trajectories are limited primarily to web-based activities and typically involve simple interactions without complex multi-step reasoning or extensive tool utilization.

**Agent Environments.** Early agent environments such as MiniWob++ (Liu et al., 2018) focused on simplified web tasks and low-level actions. Later advancements such as Mind2Web (Deng et al., 2023), WebArena (Zhou et al., 2024), and Online-Mind2Web (Xue et al., 2025) introduced more realistic websites but remained constrained in breadth and complexity. More comprehensive environments have been developed by Yao et al. (2024); Drouin et al. (2024); Xu et al. (2024) which expanded the action space and interface diversity, yet they still deviate from the actual computer environments. Recent developments like OSWorld (Xie et al., 2024) and WindowsArena (Bonatti

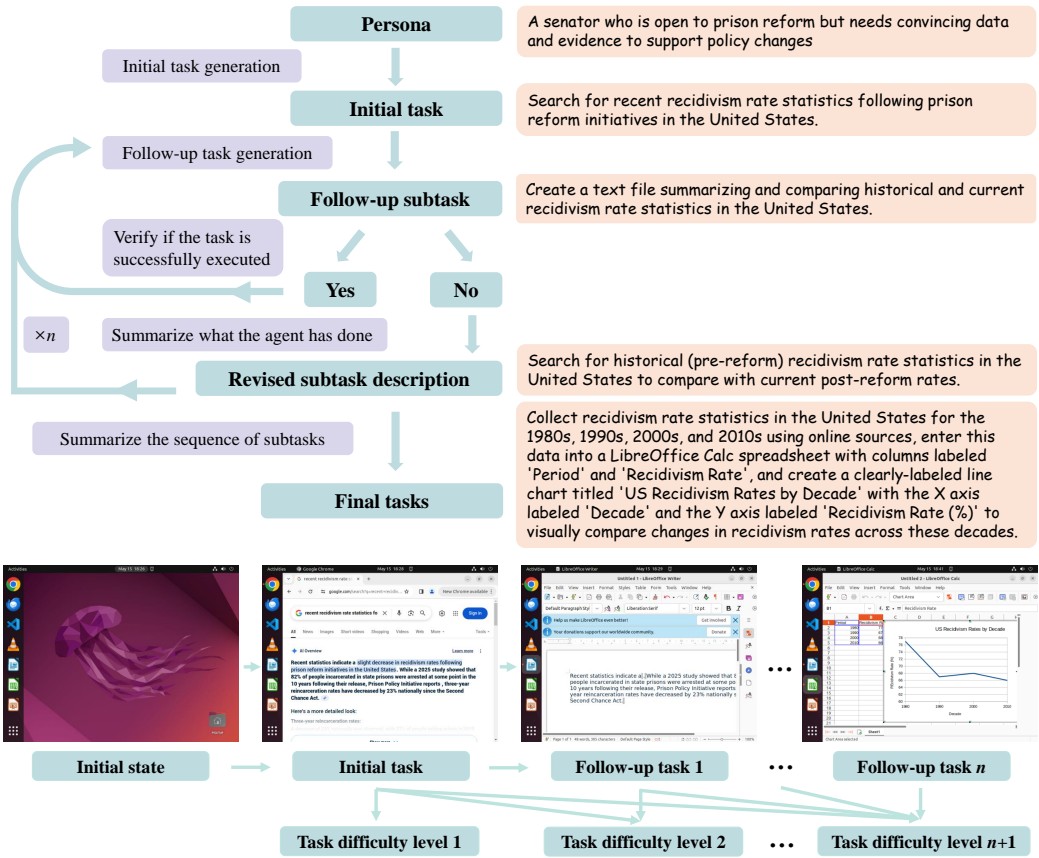

Figure 1: AgentSynth data generation pipeline. Given a persona, the task proposer generates an initial task, which is followed by a sequence of subtasks executed by the agent. Each step is verified; if execution fails, a revised subtask description is generated. After $n$ successful steps, a summarization agent composes final high-level tasks. Tasks at different difficulty levels are formed by summarizing the first 1 to $n$ subtasks, enabling controllable task complexity.

et al., 2024) address this gap by transforming real operating systems into interactive gym environments for agent training and trajectory generation. Our work leverages the capabilities of OSWorld, providing comprehensive access to authentic computer tools to enhance synthetic data generation for generalist computer-use agents.

## 3 SCALABLE AGENT TASKS AND TRAJECTORIES GENERATION

We design a synthetic data generation pipeline powered by six distinct LLM-based agents: a *task proposer*, a *task executor*, a *task verifier*, a *task reviser*, a *follow-up task proposer*, and a *task summarizer*. Central to our methodology is the exploitation of information asymmetry (Li et al., 2025); the idea that solving a task step-by-step in the forward direction is far easier than inferring the entire solution from scratch. Specifically, we generate sequences of simple, tractable subtasks, collecting trajectories along the way, and later summarizing the sequence into a single, coherent long-horizon task. This approach allows us to synthesize tasks that are easy to generate but substantially more difficult for agents to complete at test time. Full prompt templates are included in Appendix B.

Our pipeline operates in the OSWorld environment (Xie et al., 2024), a Gym-compatible simulated desktop interface that mirrors real-world computer usage. Within this environment, agents can interact freely with a broad range of software applications and system tools hosted on a virtual machine. At each step, the agent receives a full-screen screenshot (1920 × 1080), typically spanning 1k–2k tokens depending on the model's tokenizer. Based on this visual context and the current task, the LLM agent generates executable actions, such as mouse clicks, key presses, text input, and scrolling, which are executed using pyautogui (Sweigart, 2025) to closely emulate human behavior. The

full action space is detailed in Table G.1, where the percentage listed for each action type reflects its frequency of occurrence across all trajectories in our dataset. To highlight the generality of our pipeline, we also apply it to a web agent environment (InSTA (Trabucco et al., 2025)), as discussed in Appendix F. The overall data generation pipeline is detailed below and is presented in Figure 1.

**(1) Task Proposer.** We initiate the data generation process by instructing a task proposer agent to generate an initial, straightforward task. To enrich task diversity, the proposer is guided by a randomly assigned persona sampled from the persona hub (Ge et al., 2024), prompting it to suggest tasks relevant to a specific user profile. The proposer takes as input the persona and the initial Ubuntu desktop screenshot, and is prompted to create clear, specific tasks that can be completed in a few atomic actions. To ensure safety and privacy, we prohibit any tasks involving login credentials or actions such as email sending. Prompt details are provided in Appendix Table B.1.

We currently rely on GPT-4.1-based agents for task generation due to their robustness and broad generalization. Different LLM models might generate tasks with systematically different complexity, realism, or meaningfulness, and it remains an open and interesting research question how model choice affects generated task properties. Exploring task-generation variance across different LLM architectures could be beneficial to further enrich task diversity and calibrate difficulty precisely.

**(2) Task Executor.** To execute the proposed tasks, we construct a ReAct-style (Yao et al., 2023b) agent that integrates OpenAI's GPT-4.1 (OpenAI, 2025b) and computer-use-preview (OpenAI, 2025a) models. Empirically, GPT-4.1 is good at planning and interpreting visual context, while the computer-use model is more accurate in grounding actions to pixel-level coordinates. We therefore assign GPT-4.1 the role of planner: it receives the task, current screenshot, and execution history, and outputs a natural language description of the next action. This description, along with the screenshot, is then passed to the computer-use model, which generates the precise executable action (e.g., mouse click coordinates, keystrokes). This two-stage setup balances high-level reasoning with fine-grained visual grounding. During execution, we log both the model's reasoning trace and the resulting actions, enabling rich trajectory annotation. Each task execution is limited to a maximum of 10 steps. The prompts for the task executor are shown in Appendix Table B.2 and Table B.3.

**(3) Task Verifier.** The task verification agent evaluates whether a given trajectory successfully completes the intended task. It reviews the full screenshot sequence and task description, and outputs both a binary success label and a completion percentage. To avoid overwhelming the verifier with excessive visual input, we adopt a WebJudge-style architecture inspired by Xue et al. (2025). The verifier first extracts key requirements from the task description, then analyzes each screenshot to select a subset of key screenshots most relevant to task completion. The final verdict is made based on the task description, identified key requirements, and the filtered key screenshots. To reduce token usage, all screenshots are downsampled to 960×480. If a task is not fully completed, the verifier estimates the percentage of task completion. In such cases, the task reviser generates a revised task description that reflects the actual progress. Prompt details for the verifier are provided in Appendix Table B.5 to Table B.7.

**(4) Task Reviser.** When a trajectory is only partially successful, we invoke a task reviser agent to generate a revised task description that accurately reflects the actions actually completed by the agent. The reviser takes as input the full execution screenshots and identifies the goals that were successfully accomplished. It then outputs a revised task description that aligns with the observed behavior. Prompt details for the task reviser are shown in Appendix Table B.8.

**(5) Follow-up Task Proposer.** Upon completing a task, the follow-up task proposer generates the next logical subtask to continue the sequence. This agent is given the full history of prior subtasks and the most recent desktop screenshot, and is instructed to generate a simple, specific follow-up action that builds on the previous state. Additionally, the proposer is informed of previously unsuccessful tasks, prompting it to propose simpler alternatives. Like the initial proposer, it avoids tasks that require login or unsafe actions. The resulting task is executed and verified as before, and if incomplete, a revised description is generated. This iterative generation process continues until a desired sequence length is reached. Prompt templates for the follow-up proposer are shown in Appendix Table B.4.

**(6) Task Summarizer.** Finally, the task summarizer converts a sequence of completed subtasks into a single high-level task description. This summary abstracts away step-level details while preserving the overarching objective and required actions. By varying the number of subtasks summarized, we

systematically control task difficulty: more subtasks yield longer, more complex tasks that require greater reasoning and planning. This mechanism enables us to generate tasks at multiple difficulty levels in a principled way. While each subtask may be trivial in isolation, the final composed task presents a challenging, multi-step problem for LLM agents. The summarization process is illustrated in Figure 1, and prompt details are provided in Appendix Table B.9.

## 4 DATASET ANALYSIS

### 4.1 QUALITY

To assess the quality of the generated tasks and trajectories, we conducted a manual evaluation on a random sample of 100 instances across difficulty levels (approximately 16 tasks per difficulty level) to ensure representativeness across complexity. Our evaluation focused on the feasibility and realism of the overall task, the coherence and logical flow of subtasks, their relevance to the assigned persona, and the accuracy of the verifier's assessment of the agent's trajectory. Specifically, human annotators are instructed to assess:

Table 1: Human evaluation of AgentSynth task and trajectory quality.

| Quality Metric | Yes |
| --- | --- |
| Feasibility and realism | 91% |
| Subtask coherence | 90% |
| Persona Relevance | 94% |
| Verifier Accuracy | 88% |

- Feasibility and realism: Could a real human user plausibly complete this task using standard software tools?
- Subtask coherence: Does each subtask logically follow from the previous subtasks, maintaining clear and meaningful workflow progression?
- Persona relevance: Is the task aligned meaningfully with the persona provided to guide task creation?
- Verifier accuracy: Does the automated verifier's binary assessment (task success or failure) align correctly with human judgment?

As shown in Table 1, all quality metrics exceed 85%, highlighting the consistency, realism, and reliability of the data produced by the AgentSynth pipeline. On verifier accuracy, the evaluators who independently evaluated the random sample also had an inter-rater agreement of 0.74 (Cohen's kappa). We note that prior findings (Lù et al., 2025) indicate potential limitations of LLM-based verification, and our high manual-validation rate suggests our engineered verification pipeline, including selective screenshot sampling, visual context filtering, and task requirement extraction, improves verifier reliability compared to simpler methods.

### 4.2 VERIFIER CALIBRATION

We further calibrate our LLM-based verifier against human judgments on a stratified sample of trajectories. For each trajectory, humans label success/failure and assign a graded completion score in [0,1].

Figure 2 (left) reports binary agreement as a function of difficulty level. Accuracy remains high across all levels with only a mild decline as tasks

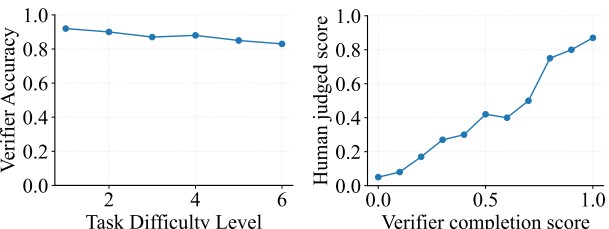

Figure 2: Verifier calibration.

become harder, indicating that the verifier's pass/fail decisions are consistent with human judgments.

Figure 2 (right) examines the verifier's completion score. We bin trajectories by verifier score and plot the average human-judged completion score in each bin. The curve is monotone: higher verifier scores correspond to higher human scores. Together, these results show that the LLM verifier is both reliable across difficulty levels and provides a meaningful partial-credit signal.

To further probe robustness, we perform an adversarial stress test of the LLM verifier. Starting from human-verified successful trajectories, we construct two types of perturbations: near-miss variants that subtly violate the goal (e.g., saving a file with an almost-correct name or in

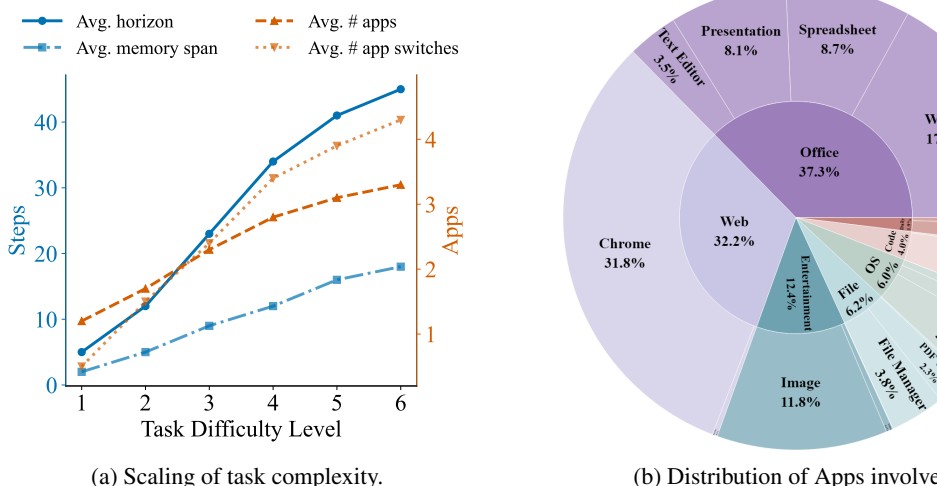

(a) Scaling of task complexity.  (b) Distribution of Apps involved.

Figure 3: AgentSynth dataset statistics.

a wrong but visually similar folder), which should be labeled as failures, and benign variants that preserve the goal but change the UI in irrelevant ways (e.g., resized windows, extra tabs), which should still be labeled as successes. Table 2 reports the fraction of perturbed states that the verifier marks as "success." The verifier incorrectly accepts only 12% of near-miss variants while correctly accepting 94% of benign variants, indicating a low false-positive rate on subtle failures and robustness to superficial UI changes.

Table 2: Adversarial stress test of verifier.

| Category | Verifier "success" |
|---|---|
| Near-miss (should fail) | 12% |
| Benign (should succeed) | 94% |

### 4.3 COMPARISON TO OTHER DATASETS AND BENCHMARKS

We designed the AgentSynth pipeline with a focus on generating diverse, realistic, and challenging data for training and evaluating computer-use agents. Table 4 compares our dataset to several existing agent benchmarks, highlighting key advantages in diversity, complexity, and scalability. Examples of tasks are shown in Appendix D.

**Diverse Real-World Tasks.** AgentSynth spans a broad range of software applications and domains, including office productivity, information retrieval, entertainment, coding, and research. This breadth ensures rich task diversity and supports generalization across practical, everyday scenarios. The pipeline leverages versatile environments that require agents to fluidly interact with multiple software tools within a single task. Figure 3b shows the coverage across domains and tools, illustrating the dataset's alignment with real-world complexity.

Importantly, our pipeline encourages multi-tool usage through chained subtasks. As shown in Appendix Figure C.1c, over 60% of trajectories involve two or more software applications, and more than 40% involve three or more, demonstrating the inherent compositionality of AgentSynth tasks.

**Long-Horizon Trajectories:** Real-world tasks often require extended sequences of actions involving planning, memory, and interface coordination. AgentSynth explicitly supports such long-horizon tasks by composing them from interdependent subtasks. As shown in Figure 3a, tasks at difficulty level 6 typically require 40-60 steps, exceeding the trajectory lengths of existing benchmarks. These tasks challenge agents to maintain context, manage interleaved goals, and execute multi-step plans, closely reflecting the demands of real-world computer use.

**Scaling of Task Complexity.** Although we define difficulty primarily via horizon, the tasks become harder along several complementary axes. Figure 3a shows that from level 1 to 6, the average horizon increases from 5 to 45 steps and the memory span from 2 to 18 steps, indicating longer-range dependencies. Moreover, the average number of applications rises from 1.2 to 3.3 and app

Table 3: Comparison of the cost of AgentSynth versus human annotations.

| Framework | Typical Steps | Human Hours per Task | Cost per Task |
|---|---|---|---|
| $\tau$-bench (Yao et al., 2024) | 20 - 30 | 2 | \$4 - \$50 |
| OSWorld (Xie et al., 2024) | 10 - 15 | 4.4 | \$8.8 - \$110 |
| TheAgentCompany (Xu et al., 2024) | 30 - 40 | 17 | \$34 - \$425 |
| **AgentSynth** | 40 - 60 | NA | \$0.6 |

Table 4: Comparison of AgentSynth to some existing LLM agent datasets and benchmarks.

| Framework | Multi-domain[1] | Domain Categories | Scalable[2] | Long Horizon |
|---|---|---|---|---|
| Mind2Web (Deng et al., 2023) | ✗ | Web | ✗ | ✗ |
| Online-Mind2Web (Xue et al., 2025) | ✗ | Web | ✗ | ✗ |
| WebArena (Zhou et al., 2024) | ✗ | Web | ✗ | ✗ |
| VisualWebArena (Koh et al., 2024) | ✗ | Web | ✗ | ✗ |
| INSTA (Trabucco et al., 2025) | ✗ | Web | ✓ | ✗ |
| AgentTrek (Xu et al., 2025b) | ✗ | Web | ✓ | ✗ |
| Explorer (Pahuja et al., 2025) | ✗ | Web | ✓ | ✓ |
| SWE-bench (Jimenez et al., 2024) | ✗ | Coding | ✗ | ✗ |
| WorkArena (Drouin et al., 2024) | ✓ | Enterprise Software | ✗ | ✓ |
| OSWorld (Xie et al., 2024) | ✓ | OS, Web, Office, Coding | ✗ | ✗ |
| WindowsAgentArena (Bonatti et al., 2024) | ✓ | OS, Web, Office, Coding | ✗ | ✗ |
| $\tau$-bench (Yao et al., 2024) | ✓ | Retail, Airline | ✗ | ✓ |
| TheAgentCompany (Xu et al., 2024) | ✓ | SWE, HR, Admin, PM, Research, | ✗ | ✓ |
| **AgentSynth** | ✓ | Web, OS, Office, Coding, Research | ✓ | ✓ |

switches from 0.5 to 4.3, reflecting increased context switching and cross-application coordination. Thus, higher difficulty levels systematically combine longer horizons with more complex memory and interrupt-handling demands. Additional dataset statistics and task properties are provided in Appendix C.

## 4.4 COST ANALYSIS

Beyond diversity and high quality, our data generation pipeline is also highly scalable and cost-efficient. Our approach achieves a cost of \$0.6 per trajectory with 5 follow-up subtasks. This is comparable with recent methods such as AgentTrek (Xu et al., 2025b) (\$0.55 per trajectory), Explorer (Pahuja et al., 2025) (\$0.28 per trajectory), and InSTA (Trabucco et al., 2025) (\$0.27 per trajectory). Furthermore, our method is much cheaper than human annotations for complex tasks with long trajectories. Table 3 shows the cost of several datasets from human annotations, where we assume the labor rate is in the range of \$2 - \$25 per hour. The detailed calculation of our cost and human labor hours is shown in Appendix E.

## 5 RESULTS AND DISCUSSION

### 5.1 EVALUATION SETUP

To assess the general-purpose computer-use capabilities of current language models, we evaluated several state-of-the-art multimodal agents with visual understanding. At each interaction step, the model receives a prompt containing the task description, the current desktop screenshot, and its own previous thoughts. The model is then asked to generate executable Python code using the `pyautogui` library to perform the next action. We sampled 50 tasks from each difficulty level for agent evaluation. Additionally, to benchmark human performance, we evaluated 20 tasks sampled from difficulty level 6, the most challenging tier in AgentSynth.

---

[1] We use "multi-domain" to indicate coverage across different types of software environments (e.g., OS tools, native applications, web interfaces), rather than just topic categories within a single interface modality like websites. While WebArena and Mind2Web span many topics, their environments are restricted to web browsers.

[2] "Scalable" indicates that the dataset can be expanded via automated or synthetic pipelines without requiring additional manual annotation.

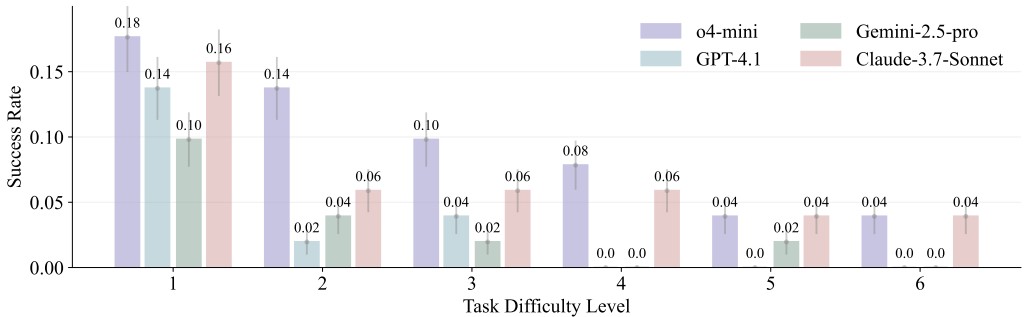

Figure 4: Model performance across task difficulty levels.

To isolate the role of the underlying language model and focus on the task difficulty itself, we use bare LLMs without fine-tuning or additional agent-specific scaffolding. Each model is prompted to generate `pyautogui` actions step-by-step based on the screenshot, task description, and action histories. This setup reflects a lower bound on performance and is intended to benchmark agents under minimal guidance rather than deploy optimized, production-grade agents. Prompts used for evaluation are detailed in Appendix B.10. Task completion is assessed using the automatic verifier agent introduced in section 3, which analyzes the full trajectory and determines whether the task was successfully completed.

## 5.2 RESULTS

The top panel of Figure 4 shows the success rates of four state-of-the-art language models on the AgentSynth benchmark across task difficulty levels 1 through 6. Despite having visual capabilities and strong general reasoning skills, all models exhibit poor performance on our benchmark, especially as task complexity increases. In contrast, humans achieve a 70% success rate even on the most difficult tasks, underscoring the performance gap. Key observations include:

**Sharp Decline with Difficulty.** All models show a consistent and steep drop in success rate as task difficulty increases. For example, `o4-mini` achieves 18% success on level 1 but drops to 4% by levels 5 and 6. `GPT-4.1` fails to complete tasks beyond level 3. This highlights the significant challenge in realistic GUI environments and demonstrates the increasing challenge posed by longer-horizon, multi-step tasks in AgentSynth.

**Near-Zero Success on Hard Tasks.** At levels 4 and 6, only o4-mini and Claude-3.7 achieve non-zero scores, and even then, the success rate is only around 4%. This indicates that current models are far from achieving generalizable competence in realistic multi-step computer tasks, showcasing the difficulty and discriminative power of our benchmark. The results highlight the need for models that can handle long-term dependencies, maintain state, and ground their decisions in visual observations over extended sequences.

## 5.3 COMMON AGENT FAILURE MODES

Despite the promising capabilities of LLM agents, their performance on the AgentSynth benchmark remains low, with most tasks ending in failure. We identify several recurring failure modes that highlight key limitations and suggest directions for future improvement:

**Inaccurate Mouse Clicks.** A frequent failure involves imprecise mouse click coordinates. While the agent often identifies the correct UI element conceptually (e.g., the "Save" button or a browser tab), it fails to locate it precisely on screen. This results in misclicks, unintended interactions (e.g., clicking ads or wrong icons), and cascading errors, such as obscuring or losing focus on the target window. Moreover, agents often repeat the same incorrect click multiple times without adapting.

**Poor Screenshot Understanding and State Tracking.** Agents frequently fail to properly interpret the visual information in screenshots. They may misidentify popups, ads, or irrelevant overlays as part of the main task UI. Other papers benchmarking LLM agents have also found problems with perceptual grounding (Xie et al., 2024; Koh et al., 2024). This weak perceptual grounding results in repetitive or irrational actions: for example, repeatedly trying to save a file that has already been

saved. Moreover, agents often lose track of what has already been done, lacking persistent memory or state awareness.

**Lack of Recovery from Errors:** Once an agent becomes stuck, it struggles to recover. Rather than exploring alternative actions or reasoning about potential mistakes, the agent tends to repeat the same failed behavior. This lack of introspection and self-correction severely limits task completion, especially for multi-step tasks requiring contingency handling. This is a common error found with complex, long-horizon across many domains in the literature, from computer-use to general remote tasks (Xu et al., 2024; Drouin et al., 2024) to math and reasoning questions (Huang et al., 2024) among others.

## 5.4 AGENT SCAFFOLDING EVALUATION

Our primary results so far used bare LLM agents that directly map from the current screen and instruction to the next GUI action. To better match contemporary agentic systems, we additionally evaluate the Agent S3 scaffold (Gonzalez-Pumariega et al., 2025), which augments the backbone model with explicit planning, tool-aware reasoning, and self-verification over intermediate steps. Figure 5 compares success rates across difficulty levels for four backbones. For the weaker models (`o4-mini`, `GPT-4.1`), adding the S3 scaffold yields substantial gains: success roughly doubles on level 1 and remains consistently higher than the bare agents across levels. Stronger backbones (`GPT-5`, `GPT-5.1`) under S3 achieve the highest absolute performance, confirming that both model quality and scaffolding contribute to success. However, for all

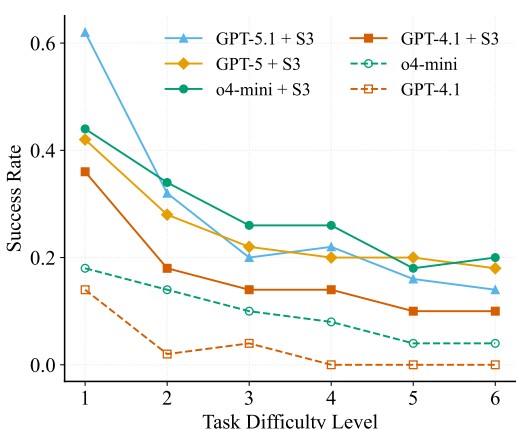

Figure 5: Model performance for bare LLMs and with Agent S3 scaffolding.

backbones the success rate still decreases as the difficulty level increases, and even the best configuration attains only modest success on level 6. This pattern indicates that while sophisticated scaffolds can recover a significant fraction of "easy" tasks, AgentSynth remains challenging for state-of-the-art computer-use agents on the hardest levels.

We also evaluate an intermediate scaffold, Agent S2.5 (Agashe et al., 2025), which includes planning and tool formatting but uses a simpler self-check mechanism. As shown in Appendix Figure C.2, Agent S2.5 yields improvements over the bare agents that are similar to Agent S3, and exhibits a similar decline in success with increasing difficulty. This suggests the qualitative trends are robust to different scaffold designs.

## 5.5 EFFECT OF INFORMATION ASYMMETRY

To assess the impact of our asymmetric generation pipeline, we compare AgentSynth to a direct-instruction baseline. In the direct setting, we prompt the same base model to generate a single long-horizon instruction in one shot, targeting approximately 5-10, 10-20, and 20-30 actions, which we denote as Easy, Medium, and Hard tasks. We then attempt to execute each instruction in OSWorld with the same executor and verifier used in our main pipeline, and retain only tasks for which a complete ground-truth trajectory can be generated. In contrast, AgentSynth first generates and verifies short subtasks and only afterwards summarizes them into long-horizon instructions.

Table 5 reports both the *generation* success rate (fraction of tasks for which a ground-truth trajectory can be obtained) and the *evaluation* success rate of Agent S3 with `GPT-5.1` on the resulting tasks. For direct instruction, generation success drops sharply as we target harder tasks: from 64% at Easy to only 11% at Hard, indicating that the generator frequently fails to produce trajectories for genuinely difficult tasks. However, among the small subset of tasks that are solvable, evaluation success remains relatively high and flat, suggesting that these retained tasks are still comparatively easy for the evaluation agent.

Table 5: Asymmetry ablation: direct instruction vs. AgentSynth.

| | Direct instruction | | | AgentSynth | | |
|---|---|---|---|---|---|---|
| | Easy | Medium | Hard | Level 1 | Level 3 | Level 6 |
| Generation success rate (%) | 64 | 33 | 11 | 65 | 57 | 52 |
| Evaluation success rate (%) | 60 | 51 | 48 | 62 | 20 | 14 |

Table 6: Effect of the base generator on evaluation performance.

| Gen. model | Eval. model | Success rate (%) at difficulty level | | | | | |
|---|---|---|---|---|---|---|---|
| | | 1 | 2 | 3 | 4 | 5 | 6 |
| GPT-4.1 | GPT-4.1 + S3 | 36 | 18 | 14 | 14 | 10 | 10 |
| | GPT-5.1 + S3 | 62 | 32 | 20 | 22 | 16 | 14 |
| GPT-5-mini | GPT-4.1 + S3 | 34 | 16 | 16 | 14 | 10 | 12 |
| | GPT-5.1 + S3 | 60 | 34 | 18 | 20 | 18 | 16 |
| GPT-5 | GPT-4.1 + S3 | 28 | 12 | 14 | 10 | 8 | 6 |
| | GPT-5.1 + S3 | 58 | 28 | 18 | 16 | 14 | 14 |
| GPT-5.1 | GPT-4.1 + S3 | 30 | 14 | 12 | 12 | 10 | 8 |
| | GPT-5.1 + S3 | 56 | 30 | 16 | 18 | 14 | 12 |

AgentSynth exhibits the opposite pattern. Generation success remains high and stable across difficulty levels, confirming that the asymmetric pipeline makes trajectory collection substantially easier. At the same time, evaluation success drops dramatically as we increase the number of summarized subtasks, from 62% at level 1 to 14% at level 6. This decoupling between generation and evaluation difficulty is precisely the effect we seek: information asymmetry allows the generator to reliably solve and log simple multi-step workflows, while the resulting composite tasks remain challenging for downstream agents.

### 5.6  EFFECT OF BASE GENERATOR

Our main experiments so far used `GPT-4.1` as the base model for proposing and executing subtasks during data synthesis. To assess how sensitive AgentSynth is to this choice, we re-run the full pipeline with three different base models: `GPT-5-mini`, `GPT-5`, and `GPT-5.1`, keeping prompts and the pipeline fixed. We then evaluate the resulting datasets using Agent S3 framework with either `GPT-4.1` or `GPT-5.1` as the backbone. Table 6 reports success rates across difficulty levels for each combination. We observe that the overall behavior is stable across generation base models. Within each dataset, the stronger evaluation model (`GPT-5.1 + S3`) consistently outperforms `GPT-4.1 + S3`, and in all cases success rates decrease as task difficulty increases. Changing the base generator from `GPT-4.1` to `GPT-5-mini` yields very similar results, while using `GPT-5` or `GPT-5.1` for synthesis leads to slightly lower success rates, suggesting that stronger generators may produce harder or more diverse tasks. These results remain consistent with the previous conclusions qualitatively. AgentSynth remains challenging at high difficulty for various generator choices, and the relative ranking of evaluation models is preserved. This indicates that our findings are not specific to a particular base model and that AgentSynth is robust to reasonable variations in the synthesis backbone.

## 6  CONCLUSION

In this work, we introduce **AgentSynth**, a scalable pipeline for synthesizing diverse, high-quality datasets of computer-use tasks and trajectories. By leveraging information asymmetry and LLM-based agents, our method decomposes complex tasks into solvable subtasks, enabling fine-grained control over difficulty and long-horizon planning. The resulting benchmark reveals performance gaps in state-of-the-art models, with success rates dropping sharply as task complexity increases, highlighting both the challenging and discriminative power of our dataset. Notably, AgentSynth can continuously and flexibly generate harder tasks, ensuring long-term benchmarking utility.

ETHICS STATEMENT

We acknowledge that we adhere to the ICLR Code of Ethics. Our data synthesis pipeline is intended solely for academic research on GUI-based computer-use agents. To ensure ethical integrity and mitigate potential risks, we explicitly prompt our proposer agents to avoid any tasks involving login credentials or real personal information, or sending forms that change the backend of any website (emails, transactions, etc). Additionally, we introduce time delays between each action to prevent excessive requests and reduce potential load on external websites, thereby minimizing any unintended impact on online services. Finally, only secure applications and websites are visited so that users of our dataset do not run into malware or other security issues. We also recognize that our method could be used to improve LLMs' abilities in harmful tasks like cyberattacks.

REPRODUCIBILITY STATEMENT

We took several steps to facilitate reproduction of our results. The end-to-end pipeline, agent roles, and OSWorld environment setup are described in Section 3, including implementation details such as screenshot resolution, action execution, and verification flow. Complete prompt templates for every component and for the evaluation agent are provided in Appendix B , and the OS-level action space used by agents is specified in Appendix G. Dataset construction, difficulty formation, statistics, and our evaluation protocol that support the main experiments are reported in Section 5. We also document the cost accounting required to replicate our scale estimates in Appendix E. We include anonymized source code and a minimal example dataset in the supplementary materials, and we will release both the full codebase and dataset publicly after the double-blind review process.

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

## A  LLMs Usage Statement

We used LLMs as general-purpose editorial assistants for this manuscript (grammar fixes, wording suggestions, and minor style edits). LLMs did not contribute to research ideation, model or algorithm design, implementation, data generation or processing, experimental setup, analysis, or results. All technical content was authored by the authors, who reviewed and approved any edits.

## B  Prompt Details

The prompts for the task proposal, execution, verification, revision, follow-up proposal, and summarization agents are provided in Tables B.1–B.9.

Table B.1: Prompt for Task Proposal Agent.

| System Role | What does this screen show? You are a real user on this computer. Please provide a single task that the user might perform on this computer and the corresponding first action towards completing that task. You can use any software in the computer and the web. Be creative and come up with diverse tasks. The task should be simple enough that can be finished in a few atomic actions. The task should be clear and very specific. |
|---|---|
| | **Task proposal rules:** |
| | 1. The task should be specific and clear. |
| | 2. The task should be achievable within 5 atomic actions like clike, scroll, type, press, etc. |
| | 3. The task should be relevant to the content of the webpage. |
| | 4. You should only propose tasks that do not require login to execute the task. |
| | 5. Provide concrete information or constraints to the task, and use mock-up information. (identifier, number, personal information, name, attributes, etc.) to make the task more specific and realistic. |
| | 7. The task description should provide all the necessary information to complete the task. |
| | **Output with JSON block:** |
| | { |
| | "thoughts":"<Detailed Thoughts and Reasons. Think about if the rules are met, e.g. why the task is related to the user character, why the task is simple enough to be finished in a few actions, is the task clear and specific, etc.>", |
| | "task":"<TASK>", |
| | "action":"<ACTION>" |
| | } |
| **User Role** | You are {PERSONA}, what task would you perform on the computer? |
| | {SCREENSHOT} |

Table B.2: Prompt for Task Execution Agent (Planner).

| | |
|---|---|
| **System Role** | You are a computer agent which perform desktop computer tasks as instructed. You have good knowledge of computer and good internet connection and assume your will controll the mouse and keyboard. For each step, you will be asked to finish a task, and you will get a screenshot of the current computer screen. You also know the actions that are already done towards the target. You need to provide the next action based on the screenshot. If you have tried similar actions several times but haven't success, analyze the reason carefully and propose a different action. Try to think if you were wrong or you missed any steps. If you think the task is finished, return "DONE" in the action field. |
| | **Rules:** |
| | 1. First analyze the screeenshot carefully, pay attention to details in the screenshot. |
| | 2. Then analyze the previous thoughts and actions to make sure you are on the right track. |
| | 3. Note: the previous actions may not be executed successfully, so you need to analyze the screenshot carefully. |
| | 4. If you find you have tried similar actions several times but haven't success, analyze the reason carefully. Try to think if you were wrong or you missed any steps. Carefully analyze the screenshot to find the reason of the failure and propose a different action. |
| | 5. If you think the task is finished, return "DONE" in the action field. |
| | **Output with JSON block:** |
| | { |
| | "thoughts":"\<Detailed Thoughts and Reasons>", |
| | "action":"\<ACTION>" |
| | } |
| **User Role** | Given the task: {TASK}. You have gathered some information {INFO}. Here is your previous thinking process to complete the task {THOUGHTS_HISTORY}. Here are your previous actions tried {ACTION_HISTORY}. Here is the current screenshot. What would be the next action? |
| | {SCREENSHOT} |

Table B.3: Prompt for Task Execution Agent (Visual Grounding).

| | |
|---|---|
| **System Role** | You are a computer agent which perform desktop computer tasks as instructed. You have good knowledge of computer and good internet connection, and assume you control the mouse and keyboard. For each step, you will be asked to finish a task, and you will get a screenshot of the current computer screen. You also know the actions that are already done towards the target. You need to provide the next action based on the screenshot. |
| | **Rules:** |
| | 1. You have all the permissions to proceed, and you don't need to ask for permission. The safety checks are acknowledged, and you can always proceed. |
| | 2. If you have clicked the same item several times, you don't need to click it again. |
| | 3. Do not click on ads. |
| | 4. If the computer is locked, type "password". |
| **User Role** | Given the task: {TASK}, you have done the following actions: {ACTION_HISTORY}. You need to do the next step: {STEP}. What would be the action? |
| | {SCREENSHOT} |

Table B.4: Prompt for Follow-up Task Proposal Agent.

| System Role | What does this screen show? You are a real user on this computer. Given the tasks the user has done, please provide a single follow-up task that the user might perform on this computer and the corresponding first action towards completing that task. You can use any software on the computer and the web. Be creative and come up with diverse tasks. The task should be simple enough that can be finished in a few atomic actions. |
|---|---|
| | **Task proposal rules:**
1. The task should depend on the previous tasks.
2. The task should be achievable within 5 atomic actions like click, scroll, type, press, etc.
4. The task should be relevant to the content of the previous tasks.
5. You should only propose tasks that do not require a login to execute the task.
6. Provide concrete information or constraints to the task, and use mock-up information (identifier, number, personal information, name, attributes, etc.) to make the task more specific and realistic.
7. The task description should provide all the necessary information to complete the task.
8. Do not propose tasks including sending emails or sharing on social media. |
| | **Output with JSON block:**
{
"thoughts":"<Detailed Thoughts and Reasons. Think about if the rules are met, e.g. why the task is related to the user character and previous tasks, why the task is simple enough to be finished in a few actions, is the task clear and specific, etc.>",
"task":"<TASK>",
"action":"<ACTION>"
} |
| User Role | You are {PERSONA}. Given the task history {TASK_HISTORY}, what would be a follow-up task? Note that these tasks {FAILED_TASKS} are too hard for the agent, propose a simpler one.
{SCREENSHOT} |

Table B.5: Prompt for Task Verification Agent (Key Requirement Identification).

| System Role | You are an expert tasked with analyzing a given task to identify the key points explicitly stated in the task description. Carefully analyze the task description and extract the critical elements explicitly mentioned in the task for achieving its goal. |
|---|---|
| | **Rules**
1. Read the task description carefully.
2. Identify and extract key points directly stated in the task description.
3. A key point is a critical element, condition, or step explicitly mentioned in the task description.
4. Do not infer or add any unstated elements. |
| | **Output with JSON block:**
{
"thoughts":"<Your detailed thinking and reasoning process>",
"key_points":"<List of the key points for completing this task>"
} |
| User Role | Given the task {TASK}, what are the key points? |

Table B.6: Prompt for Task Verification Agent (Key Screenshot Identification).

| | |
|---|---|
| **System Role** | You are an expert evaluator tasked with determining whether a screenshot contains information about the necessary steps to complete a task. Analyze the provided image and decide if it shows essential steps or evidence required for completing the task. Use your reasoning to explain your decision.

**Rules**
1. Provide a detailed description of the screenshot, including its contents, visible elements, text (if any), and any notable features.
2. Carefully examine the screenshot and evaluate whether it contains necessary steps or evidence crucial to task completion.
3. Identify key points that could be relevant to task completion, such as actions, progress indicators, tool usage, applied filters, or step-by-step instructions.
4. Does the screenshot show actions, progress indicators, or critical information directly related to completing the task
5. Is this information indispensable for understanding or ensuring task success?
6. If the screenshot contains partial but relevant information, consider its usefulness rather than dismissing it outright.
**Output with JSON block**:
{
"thoughts":"<Your detailed thinking and reasoning process>",
"necessary":"<True or False>"
} |
| **User Role** | Given the task {TASK}, the key points to finish the task {KEY_POINTS}, and the screenshot of an action, is this screenshot a necessary step to complete the task?
{SCREENSHOT} |

Table B.7: Prompt for Task Verification Agent (Final Judgement).

| | |
|---|---|
| **System Role** | You are an expert in evaluating the performance of a computer-use agent. The agent is designed to help a human user complete a computer-use task. Given the user's task, the agent's action history, key points for task completion, and some important screenshots in the agent's trajectory, your goal is to determine whether the agent has completed the task and achieved all requirements.

**Rules**
1. The filtered results must be displayed correctly. If filters were not properly applied (i.e., missing selection, missing confirmation, or no visible effect in results), it should be considered a failure.
2. You must carefully check whether these screenshots and action history meet these key points. Ensure that specific requirements are correctly applied. 3. Some tasks require a submission action or a display of results to be considered successful. Repeat actions or actions that do not lead to a visible result should be considered a failure.
4. If the agent loops through a sequence of actions that do not make progress toward the goal (including failing to click "Save" or "Submit," etc.), it should be considered a failure.

**Output with JSON block**:
{
"thoughts":"<Your detailed thinking and reasoning process>",
"success":"<True or False>",
"success rate":"<Probability of success in unit of percentage, like 20, 50, 100, numbers only>" } |
| **User Role** | Given the task {TASK}, the key points to finish the task {KEY_POINTS}, and the screenshots history, is the agent successful?
{LIST OF SCREENSHOTS} |

Table B.8: Prompt for Task Revision Agent.

| System Role | Given a list of screenshots of actions performed on the computer, you are asked to come up with a single task description that will be accomplished by performing these actions in the given sequence. First analyze these actions and generate the task description. Only summarize the completed actions, and ignore the actions that are not completed. If there is no completed action or no meaningful task, return "NONE" in the task field. |
|---|---|
| | **Rules:**
1. The task should be specific and clear.
2. The task description should provide all the necessary information to complete the task.
3. The task should be feasible to complete by a real user and should not require any additional information that is not specified in this input. |
| | **Output with JSON block:**
{
"thoughts":"<Detailed Thoughts and Reasons>",
"task":"<TASK>"
} |
| User Role | Given the set of screenshots of actions, what would be a single task description that will be accomplished by performing these actions in the given sequence?"
{LIST OF SCREENSHOTS} |

Table B.9: Prompt for Task Summarize Agent.

| System Role | Given a list of subtasks performed on the computer, you are asked to come up with a single task description that will be accomplished by performing these subtasks in the given sequence. First analyze these subtasks and generate the task description. |
|---|---|
| | **Rules**:
1. The task should be specific and clear.
2. The task description should provide all the necessary information to complete the task.
3. The task should be feasible to complete by a real user and should not require any additional information that is not specified in this input.
4. The task should include all the numbers and information used in the subtasks. |
| | **Output with JSON block**:
{
"thoughts":"<Detailed Thoughts and Reasons>",
"task":"<TASK>"
} |
| User Role | Given the subtasks history {TASK_HISTORY} and the final screenshot, what would be a single task description that will be accomplished by performing these subtasks in the given sequence?
{SCREENSHOT} |

Table B.10: Prompt for Evaluation Agent.

| | |
|---|---|
| **System Role** | You are an agent which follow my instructions and performs desktop computer tasks as instructed. You have good knowledge of computers and a good internet connection, and assume your code will run on a computer for controlling the mouse and keyboard. For each step, you will get an observation of the desktop through a screenshot. And you will predict the action of the computer based on the image and text information. |
| | You are required to use 'pyautogui' to perform the action grounded to the observation. You can use the following functions: |
| | pyautogui.click(x, y, button); 
 pyautogui.doubleClick(x, y); 
 pyautogui.moveTo(x, y); 
 pyautogui.write(string); 
 pyautogui.dragTo(x, y); 
 pyautogui.scroll(amount); 
 pyautogui.press(key); 
 pyautogui.hotkey(key1, key2, ...); 
 time.sleep(5) |
| | Note that 'pyautogui' and 'time' packages have already been imported. Return one line or multiple lines of python code to perform the action each time, be time efficient. When predicting multiple lines of code, make some small sleep like time.sleep(1). You need to to specify the coordinates of by yourself based on your observation of current observation, and you should be careful to ensure that the coordinates are correct. |
| | If you think the task is finished, return "DONE" in the code field. |
| | Output must be a valid JSON block with the following format: 
 { 
 "thoughts":"<Detailed Thoughts and Reasons>", 
 "code":"<Python code>" 
 } |
| **User Role** | Given the task: {TASK}. Here are your previous thoughts {THOUGHTS_HISTORY}. And here is the current screenshot. What would be the next action? 
 {SCREENSHOT} |

# C    ADDITIONAL DATASET STATISTICS AND EVALUATION

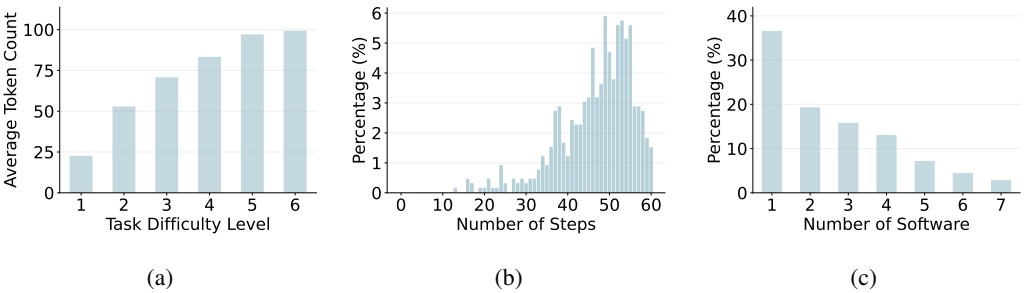

(a)  (b)  (c)

Figure C.1: AgentSynth dataset statistics. (a) Average token count by task difficulty. (b) Distribution of the number of steps at task difficulty level 6. (c) Distribution of the number of software applications for each task at difficulty level 6.

Table C.1 summarizes several task properties as a function of difficulty level. Beyond the primary "horizon" measure (average number of steps), we compute the average number of distinct applications involved, the number of application switches, and a memory-span metric defined as the maximum step distance between reading a piece of information and later using it in the same trajectory. We also estimate the fraction of tasks that require fine-grained visual discrimination ("fine-grained").

As difficulty increases from level 1 to 6, horizon, number of apps, context switches, and memory span all increase, indicating that higher levels not only involve longer trajectories but also more cross-application coordination and longer-range dependencies. The fraction of fine-grained perception tasks remains relatively stable, suggesting that visual discrimination is present across all levels but is not the primary knob we use to control difficulty.

Table C.1: Task properties across difficulty levels.

| Level | Avg. horizon | Avg. # apps | Avg. # switches | Avg. memory span | Fine-grained (%) |
|---|---|---|---|---|---|
| 1 | 5 | 1.2 | 0.5 | 2 | 40 |
| 2 | 12 | 1.7 | 1.5 | 5 | 42 |
| 3 | 23 | 2.3 | 2.4 | 9 | 38 |
| 4 | 34 | 2.8 | 3.4 | 12 | 41 |
| 5 | 41 | 3.1 | 3.9 | 16 | 45 |
| 6 | 45 | 3.3 | 4.3 | 18 | 43 |

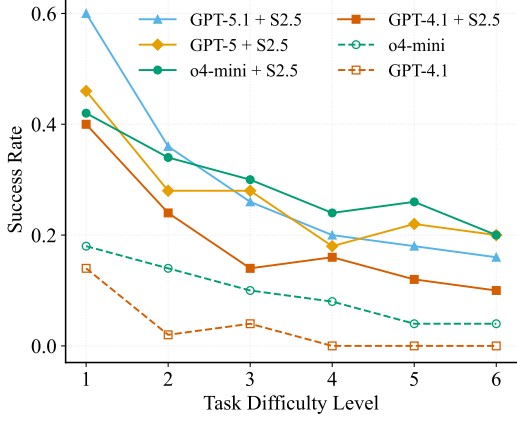

Figure C.2: Model performance for bare LLMs and with Agent S2.5 scaffolding.

## D    EXAMPLE TASKS

To illustrate the quality and realism of tasks generated by the AgentSynth pipeline, we present a representative example.

First, a persona is sampled from the persona hub (Ge et al., 2024):

> **Persona:** a senior student at Kentucky Wesleyan College.

Then, the task proposer generated an initial task tailored to this persona:

> **Initial task:** Search for the 'Kentucky Wesleyan College 2024 academic calendar' in Google Chrome.

Next, this initial task was successfully executed, and five follow-up tasks were iteratively generated and completed:

> - **Follow-up Task 1** Find the Kentucky Wesleyan College 2024 commencement (graduation) date on the academic calendar currently open in Chrome
> - **Follow-up Task 2** Open the Calendar application after searching for graduation-related dates on an academic calendar website.
> - **Follow-up Task 3** Scroll backwards month-by-month in the calendar application from March 2025 to June 2024 using the month view.
> - **Follow-up Task 4** Create a new calendar event on Kentucky Wesleyan College's 2024 commencement titled 'Graduation Day' and add a note: 'Remember to bring gown and arrive 1 hour early.
> - **Follow-up Task 5** Add a notification/reminder to the 'Graduation Day' event on May 3, 2024 in the Calendar app to alert you 1 day before.

Each of these subtasks is simple and logically follows from the previous one. The task summarizer composes them into coherent, high-level tasks. We define *task difficulty level* $n$ as the summary of the first $n$ subtasks, resulting in increasingly complex and realistic scenarios. These summarized tasks are then used for evaluation and benchmarking. The list below shows the final task descriptions at six difficulty levels, with *italicized text* indicating the incremental complexity introduced at each level:

> - **Task Difficulty Level 1** (same as the initial task): Search for the 'Kentucky Wesleyan College 2024 academic calendar' in Google Chrome.
> - **Task Difficulty Level 2**: Find and report the date and time of the Kentucky Wesleyan College 2024 commencement ceremony by searching for the 2024 academic calendar online, locating the official calendar, and *identifying the commencement event listed there*.
> - **Task Difficulty Level 3**: Search for the 'Kentucky Wesleyan College 2024 academic calendar' in Google Chrome, find the 2024 commencement date, and then *open the Calendar application to view or record the commencement date*.
> - **Task Difficulty Level 4**: Find the Kentucky Wesleyan College 2024 commencement date using Google Chrome, then open the Calendar application and *scroll back in month view from March 2025 to June 2024* in preparation for viewing or adding the graduation date to the calendar.
> - **Task Difficulty Level 5**: Find the Kentucky Wesleyan College 2024 commencement date by searching online (using the academic calendar), and *create a new event titled 'Graduation Day' in your digital Calendar application, adding a note that says 'Remember to bring gown and arrive 1 hour early'*.
> - **Task Difficulty Level 6**: Find the Kentucky Wesleyan College 2024 commencement date by searching the 2024 academic calendar online, then create a calendar event titled 'Graduation Day' in the Calendar application with a note saying 'Remember to bring gown and arrive 1 hour early,' and *set a reminder to alert you one day before the event*.

As the task level increases, both task length and complexity grow accordingly. Each additional subtask introduces new actions, tools, or planning steps. Figure C.1a shows the average token

count across task levels, confirming that longer task compositions correspond to more elaborate task descriptions and execution requirements.

We show a few more example tasks generated by our AgentSynth pipeline.

---

**Example 1**

**Persona:** A climate-conscious teenager who confronts their parent about the negative environmental impact of their investments

**Subtasks:**

- Search for a recent article about the environmental impact of investing in fossil fuel companies and open it to prepare for discussion with your parent.
- Create and edit a new text file using Visual Studio Code.
- Add a section to your text file in Visual Studio Code listing three alternative sustainable investment options (e.g., renewable energy funds, green bonds, or ESG index funds) and a brief reason why each is better for the environment.
- Add a relevant quote or statistic from the recent article about fossil fuel investments' environmental impact to your text file in Visual Studio Code, citing the source.
- Enter energy source categories ('Fossil Fuels' and 'Renewables') and their respective values (200 and 20) into a LibreOffice Calc spreadsheet, then select this data to initiate the creation of a pivot table.
- Select a range of data in LibreOffice Calc and initiate the creation of a pivot table using the selected data.

**Tasks:**

- Search for a recent article about the environmental impact of investing in fossil fuel companies and open it to prepare for discussion with your parent.
- Search for a recent article about the environmental impact of investing in fossil fuel companies, open it to gather information, and then create a new text file in Visual Studio Code where you write or summarize the key points from the article to prepare for a discussion with your parent.
- Research and summarize a recent article about the environmental impact of investing in fossil fuel companies, then use Visual Studio Code to create a new text file. In this file, document the summary and add a section listing three alternative sustainable investment options (e.g., renewable energy funds, green bonds, ESG index funds), including a brief explanation for each on why it is better for the environment, to prepare for a discussion with your parent.
- Research a recent article about the environmental impact of investing in fossil fuel companies, then create a new text file in Visual Studio Code. In the file, summarize the article by including a relevant quote or statistic (with proper citation), and list three alternative sustainable investment options, renewable energy funds, green bonds, and ESG index funds, each with a brief explanation of their environmental benefits, to prepare for a discussion with your parent.
- Prepare for a discussion about the environmental impact of investing in fossil fuel companies by (1) finding and reviewing a recent article on the topic, (2) creating a text file in Visual Studio Code summarizing three alternative sustainable investment options (such as renewable energy funds, green bonds, or ESG index funds) and explaining why each is better for the environment, (3) adding a relevant quote or statistic from the article to the text file with a citation, and (4) entering the energy source categories 'Fossil Fuels' (value 200) and 'Renewables' (value 20) into a LibreOffice Calc spreadsheet, then selecting this data to initiate the creation of a pivot table for comparison.
- Research a recent article about the environmental impact of investing in fossil fuel companies and open it. Use the information to prepare a text file in Visual Studio

---

Code that includes: (1) three alternative sustainable investment options (such as renewable energy funds, green bonds, or ESG index funds) and a brief reason each is better for the environment, and (2) a relevant quote or statistic from the article about fossil fuel investments. Then, in LibreOffice Calc, enter the values for two energy source categories: 'Fossil Fuels' (200) and 'Renewables' (20), select this data, and initiate the creation of a pivot table to compare them for a discussion with your parent

- Research a recent article about the environmental impact of investing in fossil fuel companies and open it. Use the information to prepare a text file in Visual Studio Code that includes: (1) three alternative sustainable investment options (such as renewable energy funds, green bonds, or ESG index funds) and a brief reason each is better for the environment, and (2) a relevant quote or statistic from the article about fossil fuel investments. Then, in LibreOffice Calc, enter the values for two energy source categories: 'Fossil Fuels' (200) and 'Renewables' (20), select this data, and initiate the creation of a pivot table to compare them for a discussion with your parent

---

**Example 2**

**Persona:** A mentor who guides the student in exploring more advanced concepts and experiments

**Subtasks:**

- Create a new Python file named 'experiment1.py' using VS Code to start an advanced programming experiment.
- Install the recommended 'Python' extension for Visual Studio Code to enable advanced Python features for your experiment.
- Write a simple print statement (e.g., print('Advanced experiment ready!')) in 'experiment1.py' and run the file in VS Code to confirm successful Python setup.
- Create and run a Python script ('experiment1.py') in Visual Studio Code that first prints 'Advanced experiment ready!', then edit the script to define and call a function 'greet()' that prints 'Welcome to advanced Python!', and run the updated script to confirm the new output.
- Create and test a Python function named personal_greet that takes a name as an argument and returns 'Hello, name! Welcome to advanced Python!', then call it with the name 'Alice' and print the result.
- Edit a Python script to prompt the user for their name and print a personalized greeting using a function called personal_greet.

**Tasks:**

- Create a new Python file named 'experiment1.py' using VS Code to start an advanced programming experiment.
- Set up an advanced Python programming environment in Visual Studio Code by creating a new Python file named 'experiment1.py' and installing the recommended 'Python' extension for VS Code to enable advanced Python features.
- Set up an advanced Python programming environment in Visual Studio Code by creating a new file named 'experiment1.py', installing the recommended 'Python' extension for VS Code, writing the code print('Advanced experiment ready!') in 'experiment1.py', and running the file in VS Code to confirm the Python environment is correctly configured.
- Set up Visual Studio Code for advanced Python development on your system by creating a new Python file named 'experiment1.py', installing the recommended 'Python' extension for VS Code, and confirming your environment by first adding a print statement (print('Advanced experiment ready!')) and running the file. Then,

modify 'experiment1.py' to define and call a function named 'greet()' that prints 'Welcome to advanced Python!', and run the script again to verify it outputs the new message as expected.

- Set up a Python development environment in Visual Studio Code by creating a new file named 'experiment1.py', installing the recommended Python extension, and verifying the Python setup by writing and running a script that: first, prints 'Advanced experiment ready!'; next, defines and calls a function 'greet()' that prints 'Welcome to advanced Python!'; and finally, defines and tests a function 'personal_greet' that takes a name argument and returns a personalized greeting 'Hello, name! Welcome to advanced Python!', calling it with 'Alice' and printing the result.

- Set up Visual Studio Code for advanced Python programming by installing the recommended 'Python' extension. Create a new Python file named 'experiment1.py'. In this file, first write and run a simple print statement (such as print('Advanced experiment ready!')) to confirm Python is working. Next, enhance the script by defining a function 'greet()' that prints 'Welcome to advanced Python!' and run it. Then, define a function 'personal_greet' that takes a name argument and returns a personalized message: 'Hello, name! Welcome to advanced Python!'. Call this function with the argument 'Alice' and print the output to test it. Finally, edit the script so that it prompts the user to enter their name and prints a personalized greeting using the 'personal_greet' function. Run the script after each change to verify correct output.

---

### Example 3

**Persona:** A whistleblower who anonymously provides crucial information about the art dealer's illegal operations

**Subtasks:**

- Open the web browser and navigate to the SecureDrop homepage to review the instructions for submitting anonymous tips about the art dealer's illegal operations.

- Open a text editor and draft an anonymous tip detailing the art dealer's illegal activities, including evidence (e.g., 'Art Dealer John Smith is selling forged artworks under the alias ArtHouseX. Key transaction dated 2023-09-12 with buyer code 9743. See attached invoice scan.'). Save the file as 'Anonymous_Tip_ArtDealer.txt' in the Documents folder.

- Create an invoice slide in LibreOffice Impress with the following details: Title 'Invoice', Date: 2023-09-12, Buyer Code: 9743, Seller: John Smith (Alias: ArtHouseX), Description: Forged artwork, Total: $12,000.

- Export the invoice slide as a PDF file named 'Invoice_ArtDealer_20230912.pdf' in the Documents folder for submission as evidence.

- Create and save an anonymous tip text file ('Anonymous_Tip_ArtDealer.txt') summarizing and reporting the details of a forged artwork transaction documented on an invoice, including the seller's alias, transaction date, buyer code, and a reference to the attached invoice scan.

- Write an anonymous tip about an art dealer selling forged artworks in a text file named 'Anonymous_Tip_ArtDealer.txt'.

**Tasks:**

- Open the web browser and navigate to the SecureDrop homepage to review the instructions for submitting anonymous tips about the art dealer's illegal operations.

- Review the SecureDrop homepage for instructions on anonymously submitting tips, then draft and save an anonymous tip in a text file named 'Anonymous_Tip_ArtDealer.txt' in the Documents folder. The tip should state: 'Art Dealer John Smith is selling forged artworks under the alias ArtHouseX. Key transaction dated 2023-09-12 with buyer code 9743. See attached invoice scan.

- Prepare and save an anonymous tip about art dealer John Smith's illegal activities for submission via SecureDrop by (1) reviewing SecureDrop instructions online, (2) drafting and saving a detailed anonymous tip as 'Anonymous_Tip_ArtDealer.txt' in the Documents folder, including evidence that John Smith (alias: ArtHouseX) sold forged artwork on 2023-09-12 to buyer code 9743 for $12,000, and referencing an attached invoice, and (3) creating and saving an invoice slide in LibreOffice Impress with the following details: Title 'Invoice', Date: 2023-09-12, Buyer Code: 9743, Seller: John Smith (Alias: ArtHouseX), Description: Forged artwork, Total: $12,000.

- Prepare an anonymous tip about illegal forged artwork sales by art dealer John Smith (alias: ArtHouseX), including supporting evidence. Draft an anonymous tip in a text editor stating that 'Art Dealer John Smith is selling forged artworks under the alias ArtHouseX. Key transaction dated 2023-09-12 with buyer code 9743. See attached invoice scan.' Save this draft as 'Anonymous_Tip_ArtDealer.txt' in your Documents folder. In LibreOffice Impress, create an invoice slide with the title 'Invoice', date '2023-09-12', buyer code '9743', seller 'John Smith (Alias: ArtHouseX)', description 'Forged artwork', and total '$12,000'. Export this slide as 'Invoice_ArtDealer_20230912.pdf' to the Documents folder. These two files will be prepared for anonymous submission via SecureDrop after reviewing the SecureDrop submission instructions.

- Prepare and save a complete anonymous report for submission to SecureDrop about art dealer John Smith's illegal sale of forged artworks (alias: ArtHouseX), including drafting an anonymous tip file ('Anonymous_Tip_ArtDealer.txt') outlining the activities and transaction (dated 2023-09-12, buyer code 9743, total $12,000), and creating an invoice as corroborating evidence (slide in LibreOffice Impress with these details) exported as 'Invoice_ArtDealer_20230912.pdf', both saved in the Documents folder, in accordance with the reviewed SecureDrop submission instructions.

- Prepare and save two files in the Documents folder to submit an anonymous tip about art dealer John Smith (alias ArtHouseX) selling forged artworks: (1) a text file named 'Anonymous_Tip_ArtDealer.txt' that details the illegal transaction (dated 2023-09-12, buyer code 9743, evidence reference to attached invoice scan), and (2) a PDF export of an invoice slide (created in LibreOffice Impress) titled 'Invoice', with the same transaction details (seller: John Smith (ArtHouseX), description: forged artwork, total: $12,000), saved as 'Invoice_ArtDealer_20230912.pdf', ready for anonymous submission via SecureDrop.

# E  COST ANALYSIS

In this section, we analyze the cost of AgentSynth and compare it to the labor costs of several human-curated agent datasets. Since many of these datasets are created by graduate students, a labor rate of $25 USD per hour is a reasonable baseline. However, labor rates vary widely across regions and can be as low as $2 per hour in some low-income countries. To account for this variability, we estimate costs using a range of $2–$25 per hour.

$\tau$-bench (Yao et al., 2024) includes tasks with up to 30 steps, but the authors did not provide a detailed breakdown of the human labor involved in the data curation process. The dataset construction comprises three stages: (1) manual design of the database, (2) automatic data generation, and (3) manual task annotation. Labor costs for the first two stages are not disclosed. For the third stage, the authors reported manual inspection of over 40 trials per task. Assuming an average of 3 minutes per trial, this implies approximately 2 hours of labor per task. At the estimated labor rate of $2-$25/hour, the cost per task is around $4 - $50. This should be considered a lower-bound estimate, as it does not include the potentially substantial cost of database and API design.

**OSWorld** (Xie et al., 2024) limits each task to a maximum of 15 steps and provides clear reporting on human labor. The dataset was created by 9 computer science students over approximately 1,800

person-hours, resulting in 412 tasks. This averages to about 4.4 hours per task, or $8.8 - $110 per task, assuming a labor rate of $2-$25/hour.

**TheAgentCompany** (Xu et al., 2024) reports a total of 3,000 person-hours for the creation of 175 tasks. This equates to roughly 17.1 hours of labor per task, yielding a cost of approximately $34 - $425 per task, assuming a labor rate of $2-$25/hour.

**AgentSynth.** We use GPT-4.1 ($2 per million tokens) for most agents in the pipeline, the computer-use-preview model ($3 per million tokens) for visual grounding, and GPT-4.1-mini ($0.40 per million tokens) for the verifier to reduce costs. A full-resolution screenshot (1920 ×1080) typically consumes around 1k tokens for GPT-4.1 and 2k tokens for the computer-use-preview model. For verification, we downsample screenshots to 960× 540, which results in approximately 1k tokens per image for GPT-4.1-mini. Each execution step costs roughly $0.011, and a typical trajectory consists of 50 steps across 6 subtasks. This results in an average total cost of approximately $0.60 per trajectory. Since we can generate 6 tasks from the 6 subtasks, each task is only $0.1.

# F    APPLYING AGENTSYNTH PIPELINE FOR WEB AGENT

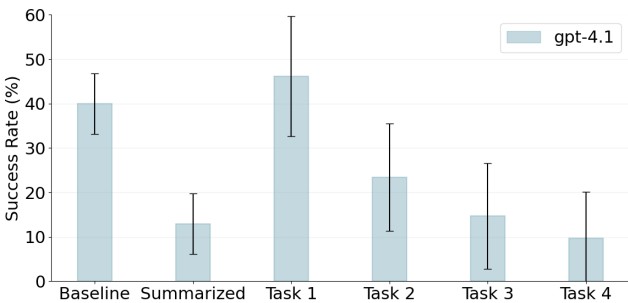

Figure F.1: Comparison of model performance at different task levels for the web agent built in the InSTA environment. 2-sigma error bars are included.

To highlight the generality of our pipeline, we also apply it to a web agent environment (InSTA (Trabucco et al., 2025)). This section describes results for our test dataset of 194 tasks in 61 task sequences from InSTA, which converts webpages into text outlines with interactable elements annotated with numerical IDs. The agent can choose from a set list of actions like clicking, filling out text fields, and going to websites, on any of the interactable elements.[3] A Playwright server hosted on Docker (Merkel, 2014) is then used to access the Internet and send GET and POST requests. At least 20GB is needed for the InSTA Docker image, and ideally more for storing Playwright calls; on an E2 Google Cloud Compute VM with 8 vCPUs, generating five task sequences using two workers takes somewhere from 60-90 minutes.

In InSTA, as shown in Figure F.1, similar drop-offs in task difficulty occur as in Figure C.1a. Summarized tasks are substantially more difficult than the baseline existing tasks in the InSTA dataset (Trabucco et al., 2025), from 40% and 12% for gpt-4.1 and o4-mini respectively to 12.9% and 6.45%. Furthermore, as shown in Figure F.2, InSTA summarized tasks also grow in description length as the number of tasks included increases.

---

[3]Since websites are less general-purpose than applications, most personas are not relevant to a given website, so personas are not used for InSTA task generation.

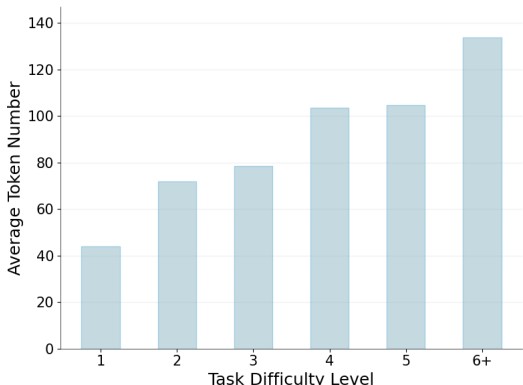

Figure F.2: Average token count for different task levels in InSTA. The tasks become more complex as the number of subtasks increases.

Table F.1: Prompt for Task Proposer Agent for InSTA.

| System Role | What does this screen show? Imagine you are a real user on this webpage. Given the webpage link, please provide a single task that a user might perform on this website and the corresponding first action towards completing that task. You can use any software in the computer and the web. Be creative and come up with diverse tasks. The task should be simple enough that can be finished in a few steps. You should propose tasks that are clear and specific. |
|---|---|
| | Task proposal rules: |
| | 1. The website should be explicitly mentioned in the task description. |
| | 2. The task should be specific, clear, and easily evaluated, not open-ended. |
| | 3. The task should be achievable within 1-3 steps. |
| | 4. The task should be relevant to the content of the webpage. |
| | 5. You should only propose tasks that do not require login to execute the task. |
| | 6. Generally try to avoid tasks that require sending emails, submitting forms, login, or other forms of communication. |
| | 7. Provide concrete information or constraints to the task, and use mock-up information (identifier, number, personal information, name, attributes, etc.) to make the task more specific and realistic. |
| | 8. The task description should provide all the necessary information to complete the task. |
| | 9. Do not propose tasks that are not possible in the Playwright API, such as downloading reports or opening and interacting with files. |
| | Output with JSON block: |
| | ```{"task":"<TASK>", "action":"<ACTION>"}``` |
| **User Role** | Given the website: {website} and webpage layout {webpage_text}, please propose a task. |

Table F.2: Prompt for Follow-up Task Proposer Agent for InSTA.

| | |
|---|---|
| **System Role** | What does this screen show? Imagine you are a real user on this webpage. Given the website link, and the tasks the user has done, please provide a single followup task that a user might perform on this website and the corresponding first action towards completing that task. You can use any software in the computer and the web. Be creative and come up with diverse tasks. The task should be simple enough that can be finished in a few steps.
Task proposal rules:
1. The website should be explicitly mentioned in the task description.
2. The task should depend on the previous tasks.
3. The task should be specific, clear, and easily evaluated, not open-ended.
4. The task should be achievable within 1-3 steps.
5. The task should be relevant to the content of the webpage.
6. You should only propose tasks that do not require login to execute the task.
7. Provide concrete information or constraints to the task, and use mock-up information (identifier, number, personal information, name, attributes, etc.) to make the task more specific and realistic.
8. The task description should provide all the necessary information to complete the task.
9. The task should be relevant to the overall task listed in the user prompt, if applicable. If the overall task is achievable within a few steps, you can simply propose the overall task as the followup task. If the overall task is already achieved, you can propose an extension of the overall task.
10. Do not propose tasks that are not possible in the Playwright API, such as downloading reports or opening and interacting with files.
11. Try to avoid tasks that require sending emails, submitting forms, login, or other forms of communication.
12. Avoid tasks that modify the backend state of the website.
Output with JSON block: ```{"task":"<TASK>", "action":"<ACTION>"}``` |
| **User Role** | Given the website: {website}, webpage layout: {webpage_text} and task history: {task_history}, please propose a followup task. [If given: "Overall task: {overall_task}."] |

## G  OSWORLD ACTION SPACE

Table G.1: Action space for the computer-use agent. The percentage indicates the relative frequency of each action type in the full AgentSynth dataset.

| Action Type | Description | Percentage |
|---|---|---|
| click $[x, y,$ button$]$ | Click at position $(x, y)$ with button of left or right. | 59.1% |
| write [text] | Type text. | 12.1% |
| press [key] | Press a single key. | 8.2% |
| scroll [amount] | Scroll up or down for amount. | 5.4% |
| move $[x, y]$ | Move the mouse to position $(x, y)$. | 5.3% |
| drag $[x_1, y_1, x_2, y_2]$ | Drag from $(x_1, y_1)$ to $(x_2, y_2)$. | 4.6% |
| hotkey [list of keys] | Press several keys at the same time | 3.0% |
| double-click $[x, y]$ | Double click at position $(x, y)$. | 1.5% |
| wait | Wait 5 seconds | 0.8% |

