# OpenReview forum: "AgentSynth: Scalable Task Generation for Generalist Computer-Use Agents"
_ICLR.cc/2026/Conference — ICLR 2026 Poster_

### Official Review · Reviewer_CsTg · 2025-10-19

**Soundness:** 2
**Presentation:** 3
**Contribution:** 3
**Rating:** 2
**Confidence:** 4

**Summary:**

This paper introduces AgentSynth, a scalable and cost-efficient pipeline for automatically synthesizing high-quality tasks and trajectory datasets for generalist computer-use agents. Motivated by information asymmetry, AgentSynth iteratively proposes and solves simple subtasks and summarizes them into a complex task. This process is also monitored by a task verifier and a task reviser. The paper conducts a comprehensive statistical analysis on the generated datasets. Experiments show that even strong multimodal models fall short in the challenging tasks generated through AgentSynth.

**Strengths:**

The main contribution of this paper is the establishment of a challenging benchmark that is more cost-effective compared to other methods. Additionally, the paper provides a detailed statistical analysis regarding the benchmark.

**Weaknesses:**

1. Despite its effectiveness, the "easy-to-hard" data synthesis approach in this paper is not entirely novel and can be observed in data synthesis across various other domains. This aspect diminishes the paper's innovativeness.

2. The paper devotes a significant amount of content to statistical analysis of the dataset. Some data cases presented in the main text are unnecessary, resulting in a limited experimental section and potentially giving the impression of a less robust study. It is recommended that the authors allocate more space to additional experimental analyses to provide readers with more insights.

3. The evaluation setting in the experimental section of this paper lacks the use of any agent scaffold, which is too simplistic and may not adequately represent the true capabilities of the model. For instance, if we were to follow the AgentSynth pipeline (breaking down complex problems into subtasks and gradually solving each subtask, equipped with a verifier and reviser), would this significantly improve the results?

**Questions:**

1. How significantly would replacing the base model for data synthesis with other models impact the quality of the synthesized data?

2. Can this pipeline be extended to domains beyond OS agents?

3. Can training with open-source models on this trajectory library achieve far superior results compared to closed-source models?

4. Will this dataset be open-sourced?

---

> ### Author Response · Authors · 2025-11-25
> **Author Response (1/5)**
>
> > **W1: Despite its effectiveness, the "easy-to-hard" data synthesis approach in this paper is not entirely novel and can be observed in data synthesis across various other domains. This aspect diminishes the paper's innovativeness.**
>
> We appreciate the reviewer’s careful reading. We agree that the high-level idea of moving from easy subtasks toward harder tasks appears in other domains, and we do not claim that generic idea as our contribution. Our novelty lies in how we instantiate this principle for GUI computer-use agents and in the resulting benchmark:
>
> - **Controllable difficulty**. We do not only “make tasks harder”, we define 6 difficulty levels whose horizon and multi-app complexity are systematically controlled and shown to correlate with agent success.
>
> - **Information-asymmetry design**. During data generation, the agent sees decomposed subtasks, intermediate screens, and verifier feedback, making each step easy. During evaluation, we hide this structure and expose only a single long-horizon instruction and initial screen. Tasks are thus deliberately easy for the generator but hard for the solver, which is different from most easy-to-hard pipelines (which primarily make learning easier).
>
> To make the role of information asymmetry more explicit, we added a new subsection (**Section 5.5 Effect of information asymmetry**) where we directly compare our pipeline to a direct-instruction baseline that generates difficult tasks in one shot. In the direct setting, we prompt the same base model (GPT-4.1) to generate a single long-horizon instruction in one shot, targeting Easy (5–10 actions), Medium (10–20), and Hard (20–30) tasks. We then attempt to execute each instruction in OSWorld and retain only tasks for which a complete ground-truth trajectory can be generated. Table 1 below summarizes both the *generation* success rate (fraction of tasks for which a ground-truth trajectory can be obtained) and the *evaluation* success rate of Agent S3 with GPT-5.1 on these tasks:
>
> **Table 1 – Asymmetry ablation: direct instruction vs. AgentSynth**
>
> |                          | Direct: Easy | Direct: Medium | Direct: Hard | AgentSynth: L1 | AgentSynth: L3 | AgentSynth: L6 |
> |--------------------------|-------------:|---------------:|-------------:|---------------:|---------------:|---------------:|
> | Generation success rate (%) | 64 | 33 | 11 | 65 | 57 | 52 |
> | Evaluation success rate (%) | 60 | 51 | 48 | 62 | 20 | 14 |
>
> For direct instruction, generation success drops from 64% (Easy) to only 11% (Hard), showing that the model often fails to produce valid ground-truth trajectories for complex tasks. However, among the small subset of tasks that are solvable, evaluation success remains high and relatively flat (60%–48%), indicating that the retained tasks are still relatively easy for agents.
>
> In contrast, AgentSynth maintains high generation success (>50%) even at the hardest level, yet evaluation success drops sharply from 62% at level 1 to 14% at level 6. This decoupling between generation feasibility and evaluation difficulty is exactly what our information-asymmetry design aims for: it makes trajectory collection easy while yielding benchmarks that remain challenging.
>
> We believe this concrete ablation clarifies the specific novelty of our approach beyond the generic notion of "easy-to-hard".

---

> ### Author Response · Authors · 2025-11-25
> **Author Response (2/5)**
>
> >**W2: The paper devotes a significant amount of content to statistical analysis of the dataset. Some data cases presented in the main text are unnecessary, resulting in a limited experimental section and potentially giving the impression of a less robust study. It is recommended that the authors allocate more space to additional experimental analyses to provide readers with more insights.**
>
> We thank the reviewer for pointing out that the original version devoted too much space to dataset statistics and a case study, which could make the experimental section appear limited. We took this feedback seriously in the revision and rebalanced the paper as follows:
>
> - **Moved the case study to the appendix**. The detailed persona-level case study that previously occupied a subsection in the main text is now in Appendix D. This freed space in the main paper to add new experiments without losing qualitative insight for interested readers.
>
> - **Added agent scaffolding evaluation (Sec. 5.4)**. We evaluated strong agent frameworks (Agent S3 and S2.5) with multiple backbones, showing that even advanced agents struggle on the hardest levels
>
> - **Added asymmetry ablation (Sec. 5.5)**. We compared our information-asymmetric AgentSynth pipeline to a direct-instruction baseline that generates difficult tasks in one shot. This confirms our information-asymmetry design leads to easier trajectory collection but harder evaluation.
>
> - **Added Base-generator ablation (Sec. 5.6)**. We replace the base model for data generation with GPT-5-mini, GPT-5, and GPT-5.1. Results show that synthesized data remains stable across different base models. Changing the base generator from GPT-4.1 to other models yields very similar success curves: success decreases with difficulty for all datasets.
>
> - **Added multi-axis task complexity analysis (Sec. 4.3)**. We analyzed task complexity along multiple axes (horizon, number of applications, context switches, memory span, and fine-grained perception). This indicates that AgentSynth’s difficulty levels align with several complementary difficulty axes beyond simply horizon.
>
> - **Added verifier calibration, partial-credit analysis, and adversarial stress test (Section 4.2)**. We calibrate the LLM verifier against human judgments on a stratified sample of trajectories and demonstrate that the verifier is both well aligned with human judgments and its completion score provides a meaningful partial-credit signal. Adversarial stress test indicates the verifier is robust to minor perturbations.
>
> Taken together, we believe these changes rebalance the paper toward a much richer experimental analysis and provide more insights for readers.

---

> ### Author Response · Authors · 2025-11-25
> **Author Response (3/5)**
>
> > **W3: The evaluation setting in the experimental section of this paper lacks the use of any agent scaffold, which is too simplistic and may not adequately represent the true capabilities of the model. For instance, if we were to follow the AgentSynth pipeline (breaking down complex problems into subtasks and gradually solving each subtask, equipped with a verifier and reviser), would this significantly improve the results?**
>
> We thank the reviewer for pointing out the importance of evaluating more advanced agents. We have now addressed it by adding advanced agent baselines. In the revised manuscript, (**Section 5.4: Agent Scaffolding Evaluation**) evaluates [Agent S3](https://arxiv.org/pdf/2510.02250) (Gonzalez-Pumariega et al., 2025) and [Agent S2.5](https://arxiv.org/pdf/2504.00906) (Agashe et al., 2025):
>
> - Agent S3 uses a plan–act–reflect loop: it first drafts a high-level plan, then executes actions step-by-step while maintaining a memory of state summaries, and periodically invokes a self-verifier to check subgoals and revise the plan.
>
> - Agent S2.5 is a lighter scaffold that includes planning and tool formatting but uses a simpler self-check mechanism.
>
> We run both scaffolds with several backbones (o4-mini, GPT-4.1, GPT-5, GPT-5.1) across all difficulty. The results are shown in Table 3 and Table 4 below:
>
> **Table 3 - Success rate (\%) vs. difficulty level with Agent S3 scaffolding.**
>
> | Model                  | L1  | L2  | L3  | L4  | L5  | L6  |
> |------------------------|-----|-----|-----|-----|-----|-----|
> | o4-mini (bare)         | 18  | 14  | 10  |  8  |  4  |  4  |
> | gpt-4.1 (bare)         | 14  |  2  |  4  |  0  |  0  |  0  |
> | o4-mini + S3     | 44  | 34  | 26  | 26  | 18  | 20  |
> | gpt-4.1 + S3     | 36  | 18  | 14  | 14  | 10  | 10  |
> | gpt-5   + S3     | 42  | 28  | 22  | 20  | 20  | 18  |
> | gpt-5.1 + S3     | 62  | 32  | 20  | 22  | 16  | 14  |
>
>
> **Table 4 - Success rate (\%) vs. difficulty level with Agent S2.5 scaffolding.**
>
> | Model                   | L1  | L2  | L3  | L4  | L5  | L6  |
> |-------------------------|-----|-----|-----|-----|-----|-----|
> | o4-mini (bare)          | 18  | 14  | 10  |  8  |  4  |  4  |
> | gpt-4.1 (bare)          | 14  |  2  |  4  |  0  |  0  |  0  |
> | o4-mini + S2.5    | 42  | 34  | 30  | 24  | 26  | 20  |
> | gpt-4.1 + S2.5    | 40  | 24  | 14  | 16  | 12  | 10  |
> | gpt-5   + S2.5    | 46  | 28  | 28  | 18  | 22  | 20  |
> | gpt-5.1 + S2.5    | 60  | 36  | 26  | 20  | 18  | 16  |
>
> These results show that:
>
> - Scaffolding substantially boosts performance over bare agents, especially on easier levels (e.g., o4-mini improves from 18% to 44% on L1 and 4% to 20% on L6 with Agent S3).
>
> - Success still drops sharply with difficulty for all scaffolds and backbones, and even the best configuration remains about 20% success on level 6.
>
> This demonstrates that AgentSynth is challenging not only for bare models but also for competitive, modern agent pipelines that perform planning, decomposition, and self-reflection.

---

> ### Author Response · Authors · 2025-11-25
> **Author Response (4/5)**
>
> > **Q1: How significantly would replacing the base model for data synthesis with other models impact the quality of the synthesized data?**
>
> We appreciate the reviewer's question about the impact of changing the base model for data synthesis. We agree that the dependence of AgentSynth on the synthesis backbone is an important question. In the revised manuscript we therefore added **Section 5.6: Effect of Base Generator**, where we re-run the entire pipeline with three additional base models for generation: GPT-5-mini, GPT-5, GPT-5.1, keeping prompts and the pipeline fixed. For each resulting dataset, we evaluate using the Agent S3 scaffold with either GPT-4.1 or GPT-5.1 as the evaluation backbone. The main results are summarized in Table 5 below:
>
> **Table 5 – Effect of the base generator on evaluation performance.**
>
> | Gen. model | Eval. model    | L1 | L2 | L3 | L4 | L5 | L6 |
> |-----------:|----------------|---|---|---|---|---|---|
> | GPT-4.1    | GPT-4.1 + S3   | 36 | 18 | 14 | 14 | 10 | 10 |
> |            | GPT-5.1 + S3   | 62 | 32 | 20 | 22 | 16 | 14 |
> | GPT-5-mini | GPT-4.1 + S3   | 34 | 16 | 16 | 14 | 10 | 12 |
> |            | GPT-5.1 + S3   | 60 | 34 | 18 | 20 | 18 | 16 |
> | GPT-5      | GPT-4.1 + S3   | 28 | 12 | 14 | 10 |  8 |  6 |
> |            | GPT-5.1 + S3   | 58 | 28 | 18 | 16 | 14 | 14 |
> | GPT-5.1    | GPT-4.1 + S3   | 30 | 14 | 12 | 12 | 10 |  8 |
> |            | GPT-5.1 + S3   | 56 | 30 | 16 | 18 | 14 | 12 |
>
> We observe that:
>
> - Overall behavior is stable across generation backbones. For every dataset, success rates generally decrease with difficulty, and GPT-5.1+S3 consistently outperforms GPT-4.1+S3.
>
> - Switching the base generator from GPT-4.1 to GPT-5-mini produces very similar curves, indicating robustness to modest backbone changes.
>
> - Using GPT-5 / GPT-5.1 for synthesis slightly lowers success rates for the same evaluation agent, suggesting stronger generators may produce harder or more diverse tasks.
>
> These findings indicate that our conclusions are not tied to a particular synthesis model. AgentSynth remains challenging at high difficulty for a range of base generators, and the relative ranking of evaluation agents is preserved.
>
>
> > **Q2: Can this pipeline be extended to domains beyond OS agents?**
>
> We appreciate the reviewer's question about the extensibility of our pipeline to other domains. Yes, the pipeline is environment-agnostic by design. The only environment-specific components is the executor (how actions are sent). To demonstrate this, we already implemented AgentSynth in a web environment (InSTA) and report the results in **Appendix F** of the paper.
>
> In this setting, we convert webpages into structured actions and apply the same information-asymmetry pipeline to generate and verify short web subtasks, then summarize them into long-horizon instructions. The success rate drops from 40% on the easiest level to 12% on the hardest level.
>
> This indicates our pipeline is extensible which also generates significantly harder web tasks with a similar difficulty drop-off pattern as in the desktop environment.
>
>
> > **Q3: Can training with open-source models on this trajectory library achieve far superior results compared to closed-source models?**
>
> We fully agree that training open-source agents on AgentSynth and comparing them to closed-source models is an exciting and important direction. However, training a competitive open-source multimodal agent with full GUI action space requires substantial compute and engineering, which is beyond the current scope but is a central focus of our ongoing work.
>
> Our goal here is to enable such studies. We will release the code and dataset so that the community can explore training open-source models and conduct compute-normalized comparisons against closed-source agents. We view this as a natural next step built on top of AgentSynth, and we are actively planning follow-up work in this direction.
>
> > **Q4: Will this dataset be open-sourced?**
>
> Yes, upon acceptance we will release the code and dataset, enabling both training-focused and evaluation-focused follow-up works.

---

> ### Author Response · Authors · 2025-11-25
> **Author Response (5/5)**
>
> We hope that these additions: new asymmetry ablations, richer experimental analyses, multi-axis complexity statistics, scaffolded agent evaluation, verifier calibration and stress tests, and base-generator ablations, address your concerns about novelty, robustness, and evaluation strength, and will lead you to view the revised manuscript as meeting the bar for acceptance.

---

> > ### Comment · Reviewer_CsTg · 2025-11-25
> >
> > Thanks for the authors' response. I have no further questions and have raised my score to 4, but the novelty of this paper prevents me from thinking it is acceptable.

---

> > > ### Author Response · Authors · 2025-11-27
> > > **Additional Clarification on Novelty**
> > >
> > > Thank you again for the helpful feedback. We would like to briefly clarify how we see the novelty of this work.
> > >
> > >
> > > **Conceptual novelty: resolving a reliability-hardness trade-off via information asymmetry.**
> > >
> > > On the conceptual side, our goal is to address a specific trade-off in synthetic agent benchmarks: long-horizon tasks that are hard enough for evaluation are often unreliable to generate trajectories for, while tasks that are easy to generate reliably tend to be simple at evaluation. Our information-asymmetry design is a principle to separate these two goals: during generation, the agent sees subtasks, intermediate states, and verifier feedback, while at evaluation, we hide this structure and show only a single long-horizon instruction. The asymmetry ablation we added shows this effect in practice: direct one-shot instructions have low generation success but flat evaluation success, whereas AgentSynth maintains high generation success while evaluation success drops with difficulty.
> > >
> > > **Benchmark / dataset novelty: a new regime of long-horizon, multi-domain computer-use tasks.**
> > >
> > > On the benchmark side, AgentSynth occupies a regime that, to our knowledge, existing datasets do not: realistic computer-use environments, multi-application workflows, an explicit difficulty axis tied to multiple complexity metrics, and long-horizon trajectories that are competitive for strong scaffolded agents. Prior work typically offers either real OS without scalable synthetic long-horizon tasks, or synthetic pipelines that are web-only and lack explicit difficulty control.
> > >
> > > In summary, our contributions are:
> > > - an information-asymmetry principle as a way to resolve the reliability-hardness trade-off in synthetic agent benchmarks
> > > - a concrete benchmark that realizes this principle and opens a challenging long-horizon, multi-tool regime.
> > >
> > > We hope this helps clarify how our contribution differs from prior datasets and pipelines.

---

### Official Review · Reviewer_jFe8 · 2025-10-23

**Soundness:** 2
**Presentation:** 3
**Contribution:** 3
**Rating:** 6
**Confidence:** 3

**Summary:**

The paper proposes AgentSynth, an automated pipeline that synthesizes long-horizon, realistic computer-use tasks by chaining LLM-generated simple subtasks and then summarizing them into a single complex instruction. The key idea is to exploit information asymmetry between generation and evaluation. Difficulty is tunable by subtask count, spanning web/OS/office/coding; benchmark shows steep agent performance decay with horizon. MLLMs perform poorly and degrade steeply with difficulty; the pipeline is also inexpensive ($0.60 per trajectory).

**Strengths:**

1. Clever use of information asymmetry to make generation easy but evaluation hard; difficulty is tunable via subtask count (d), enabling principled long-horizon benchmarks.
2. Results show a sharp success-rate drop as d increases, making the benchmark’s discriminative power evident.
3. Practicality & scale: diverse, multi-tool tasks on real desktop environments; cost-efficient generation with transparent cost accounting.

**Weaknesses:**

1. Limited causal evidence: missing ablations of asymmetry vs. direct instruction, and broader verifier calibration (agreement curves, partial-credit) across tools and difficulty.
2. Difficulty = horizon is not fair enough: other metrics like complementary axes (fine-grained perception, long-term memory, interrupt handling) would enrich difficulty control.
3. The verifier is still LLM-based and can misjudge corner cases. More human audits or adversarial stress tests for the verifier would strengthen claims.

**Questions:**

1. Can you provide asymmetry ablations and verifier calibration results to justify the difficulty and scoring?
2.  Although increasing task horizon typically enhances diversity and difficulty, it also increases training cost. Under a fixed training compute/token budget, have you evaluated whether curating higher-quality (e.g., cleaner supervision, lower noise, stronger verifiability, clearer goal decomposition) short/medium-horizon data can outperform simply lengthening horizons in terms of compute-normalized effectiveness?

---

> ### Author Response · Authors · 2025-11-25
> **Author Response (1/4)**
>
> > **W1: Limited causal evidence: missing ablations of asymmetry vs. direct instruction, and broader verifier calibration (agreement curves, partial-credit) across tools and difficulty.**
>
> We thank the reviewer for pointing out the need for more direct causal evidence. In the revised manuscript we added a new subsection (**Section 5.5 Effect of information asymmetry**), where we explicitly compare our asymmetric AgentSynth pipeline to a direct-instruction baseline. In the direct setting, we prompt the same base model (GPT-4.1) to generate a single long-horizon instruction in one shot, targeting Easy (5–10 actions), Medium (10–20), and Hard (20–30) tasks. We then attempt to execute each instruction in OSWorld and retain only tasks for which a complete ground-truth trajectory can be generated.
>
> Table 1 below summarizes both the *generation* success rate (fraction of tasks for which a ground-truth trajectory can be obtained) and the *evaluation* success rate of Agent S3 with GPT-5.1 on these tasks:
>
> **Table 1 – Asymmetry ablation: direct instruction vs. AgentSynth**
>
> |                          | Direct: Easy | Direct: Medium | Direct: Hard | AgentSynth: L1 | AgentSynth: L3 | AgentSynth: L6 |
> |--------------------------|-------------:|---------------:|-------------:|---------------:|---------------:|---------------:|
> | Generation success rate (%) | 64 | 33 | 11 | 65 | 57 | 52 |
> | Evaluation success rate (%) | 60 | 51 | 48 | 62 | 20 | 14 |
>
> For direct instruction, generation success drops from 64% (Easy) to only 11% (Hard), showing that the model often fails to produce valid ground-truth trajectories for complex tasks. However, among the small subset of tasks that are solvable, evaluation success remains high and relatively flat (60–48%), indicating that the retained tasks are still comparatively easy for the evaluation agent.
>
> In contrast, AgentSynth maintains high generation success (>50%) even at the hardest level, while evaluation success drops sharply from 62% at level 1 to 14% at level 6. This decoupling between generation feasibility and evaluation difficulty is precisely the effect we seek: information asymmetry makes it easy to collect long trajectories while keeping the resulting tasks hard for downstream agents. This ablation directly provides the causal evidence of the impact of asymmetry versus direct instruction.
>
>
> For the verifier, we conducted broader calibration and added a new subsection (**Section 4.2 Verifier Calibration**)
> We calibrate our LLM-based verifier against human judgments on a stratified sample of trajectories: humans label success/failure and assign a graded completion score in [0,1]. Table 2 below summarizes the verifier accuracy (verifier human agreement) across difficulty levels:
>
> **Table 2 – Verifier accuracy vs. difficulty level (human vs. verifier).**
>
> | Task difficulty level | 1  | 2  | 3  | 4  | 5  | 6  |
> |----------------------:|---:|---:|---:|---:|---:|---:|
> | Verifier accuracy (%) | 92 | 90 | 87 | 88 | 85 | 83 |
>
> Accuracy remains high across all levels with only a mild decline as tasks become harder, indicating that the verifier's pass/fail decisions are consistent with human labels.
>
> We additionally examined the partial-credit calibration. We bin trajectories by verifier completion score and calculate the average human-judged completion score in each bin. The results are summarized in Table 3 below:
>
> **Table 3 – Human-judged completion vs. verifier completion score.**
>
> | Verifier completion score  | 0 | 0.1 | 0.2 | 0.3 | 0.4 | 0.5 | 0.6 | 0.7 | 0.8 | 0.9 | 1 |
> |-----------------------------:|----:|----:|----:|----:|----:|----:|----:|----:|----:|----:|----:|
> | Avg. human-judged score   |  0.05  |  0.08  |  0.17  |  0.27  |  0.30  |  0.42  |  0.40  |  0.50  |  0.75  |  0.80  |  0.87  |
>
> As shown, there is an overall monotonic trend that higher verifier completion scores correspond to higher human-judged completion scores. Together with Table 2, these results indicate that the verifier is both well aligned with human judgments and its completion score provides a meaningful partial-credit signal.

---

> ### Author Response · Authors · 2025-11-25
> **Author Response (2/4)**
>
> > **W2: Difficulty = horizon is not fair enough: other metrics like complementary axes (fine-grained perception, long-term memory, interrupt handling) would enrich difficulty control.**
>
> We thank the reviewer for pointing out the need for more comprehensive difficulty control. And we agree that difficulty should not be reduced to horizon alone. In the revision we therefore added a dedicated analysis of task complexity along multiple axes (**Section 4.3: Scaling of Task Complexity**).
>
> Table 4 below summarizes several task properties as a function of difficulty level. Beyond the primary *horizon* measure (average number of steps), we compute the
> - average number of distinct applications involved in the trajectory
> - average number of application switches
> - memory-span metric defined as the average maximum step distance between reading a piece of information and later using it in the same trajectory
> - fraction of tasks that require fine-grained visual discrimination
>
> **Table 4 – Task properties across difficulty levels.**
>
> | Level | Avg. horizon | Avg. # apps | Avg. # switches | Avg. memory span | Fine-grained (%) |
> |------:|-------------:|------------:|----------------:|-----------------:|-----------------:|
> | 1     | 5            | 1.2         | 0.5             | 2                | 40 |
> | 2     | 12           | 1.7         | 1.5             | 5                | 42 |
> | 3     | 23           | 2.3         | 2.4             | 9                | 38 |
> | 4     | 34           | 2.8         | 3.4             | 12               | 41 |
> | 5     | 41           | 3.1         | 3.9             | 16               | 45 |
> | 6     | 45           | 3.3         | 4.3             | 18               | 43 |
>
>
> As difficulty increases from level 1 to 6, horizon, number of apps, context switches, and memory span all increase, indicating that higher levels not only involve longer trajectories but also more cross-application coordination and longer-range dependencies. The fraction of fine-grained perception tasks remains relatively stable, suggesting that visual discrimination is present across all levels but is not the primary knob we use to control difficulty. While we still expose horizon as the primary control knob for simplicity, this analysis shows that AgentSynth’s difficulty levels already track several complementary axes.
>
> > **W3: The verifier is still LLM-based and can misjudge corner cases. More human audits or adversarial stress tests for the verifier would strengthen claims.**
>
> We thank the reviewer for pointing out the need for more robust verifier evaluation. In addition to the human calibration above, we added an **adversarial stress test** for the verifier (**Section 4.2**). Starting from human-verified successful trajectories, we construct two types of perturbed final states:
>
> - *Near-miss* variants that subtly violate the goal (e.g., saving a file with an almost-correct name or into a wrong but visually similar folder), which should be labeled as failures.
>
> - *Benign* variants that preserve the goal but alter irrelevant UI aspects (e.g., resized windows, extra tabs), which should be labeled as successes.
>
> The verifier’s behavior on these stress tests is summarized in Table 5 below:
>
> **Table 5 – Adversarial stress test of the verifier.**
>
> | Category                     | Verifier "success" rate |
> |-----------------------------|------------------------:|
> | Near-miss (should fail)     | 12% |
> | Benign (should succeed)     | 94% |
>
> The verifier correctly rejects the vast majority of near-miss states (only 12% incorrectly marked as "success") and correctly accepts most benign variations (94% success). This demonstrates that the verifier is robust to minor perturbations and can maintain high accuracy even in adversarial settings.

---

> ### Author Response · Authors · 2025-11-25
> **Author Response (3/4)**
>
> >**Q1: Can you provide asymmetry ablations and verifier calibration results to justify the difficulty and scoring?**
>
> We appreciate the reviewer’s suggestions here. As discussed in our responses to W1 and W3 above, in the revised manuscript, we now explicitly provide both:
>
> - Asymmetry ablation comparing direct long-instruction generation to our information-asymmetric AgentSynth pipeline (**Section 5.5**)
> - Broader verifier calibration and stress tests (**Section 4.2**)
>
> > **Q2: Although increasing task horizon typically enhances diversity and difficulty, it also increases training cost. Under a fixed training compute/token budget, have you evaluated whether curating higher-quality (e.g., cleaner supervision, lower noise, stronger verifiability, clearer goal decomposition) short/medium-horizon data can outperform simply lengthening horizons in terms of compute-normalized effectiveness?**
>
> We appreciate the reviewer's question about the trade-off between task horizon and training cost. We agree that comparing high-quality short/medium-horizon data to longer-horizon tasks under a fixed compute or token budget is an important training question. Our paper, however, is primarily a data/benchmark contribution. We focus on the AgentSynth generation pipeline and the resulting long-horizon GUI benchmark, not on proposing or optimizing a particular training recipe.
>
> To probe this question lightly within our constraints, we conducted small-scale in-context learning experiments where we provided the model with short-, medium-, or long-horizon demonstration trajectories under a matched context-token budget. We did not observe consistent or substantial gains in success rate from any of these simple few-shot configurations. This suggests that naive in-context learning over a small number of demonstrations is insufficient to fully exploit the dataset, and that meaningful improvements likely require proper fine-tuning or RL training, which entails significant compute and engineering effort.
>
> A rigorous compute-normalized comparison between "more high-quality short/medium data" and "fewer long-horizon trajectories" would require training multiple agents under carefully matched token budgets, which is a substantial undertaking and orthogonal to our main contribution. We therefore view this as a valuable direction for follow-up work enabled by AgentSynth, rather than something we can fully resolve in this submission. We will release the code and dataset to facilitate such training-focused studies by the community.

---

> ### Author Response · Authors · 2025-11-25
> **Author Response (4/4)**
>
> In summary, we have focused on directly addressing your main concerns: (i) we added asymmetry ablations comparing our pipeline to a direct-instruction baseline, (ii) we performed verifier calibration against human judgments, including binary agreement curves, partial-credit analysis, and adversarial stress tests, and (iii) we analyzed difficulty along multiple axes. These additions are intended to provide the causal evidence, scoring reliability, and richer difficulty control.
>
> In addition to these targeted changes, we also expanded the experimental section in several ways that further strengthen the paper:
>
> - **Agent scaffolding evaluation (Sec. 5.4)**. We evaluated strong agent frameworks (Agent S3 and S2.5) with multiple backbones, demonstrating that even advanced agents equipped with planning and self-verification struggle on the hardest tasks.
>
> - **Base-generator ablation (Sec. 5.6)**. We replaced the base model for data generation with several different models. Results show that synthesized data remain stable across base models.
>
> We hope that these additions address your concerns about causal evidence, verifier reliability, and difficulty design, and that the strengthened experimental section will encourage a more positive assessment of the paper.

---

> > ### Comment · Reviewer_jFe8 · 2025-11-27
> >
> > Thank you very much for the detailed responses, which have given me a clearer and more comprehensive understanding of the paper. However, I believe that my current score remains appropriate.

---

### Official Review · Reviewer_kK3T · 2025-11-01

**Soundness:** 2
**Presentation:** 3
**Contribution:** 3
**Rating:** 6
**Confidence:** 3

**Summary:**

This paper proposes AgentSynth, a dataset of synthetic computer-use agent tasks in the OSWorld environment, generated using a subtask-based compositional approach. The approach leverages a loop of subtask-level proposal, execution, verification and potential subtask revision, then summarization to compose the subtasks into one coherent task. The authors show that baseline agent's performance suffers for more difficult tasks consisting of more subtasks, indicating the effectiveness of the AgentSynth in exposing gaps in agent capabilities.

**Strengths:**

- The paper tackles an important problem of testing LLM agents in complex tasks involving multiple subtasks, which is not easily controllable in existing benchmark works. The proposed approach is intuitive and cost effective for composing arbitrarily complex sequences of salient subtasks.
- The results indicate that baseline agents indeed suffer from compositional tasks.
- The paper is well written and easy to follow.

**Weaknesses:**

- My main concern with the paper is the lack of advanced agent approaches evaluated on the benchmark. The baseline agent tested consists of a basic agent with a simple prompt, and in my understanding only represents the performance lowerbound on the benchmark. Without evaluating more advanced approaches, it is difficult to understand how well competitive agents would perform on the benchmark.
- A missing citation is [1].

[1] Exposing Limitations of Language Model Agents in Sequential-Task Compositions on the Web. Furuta et al., TMLR 2024

**Questions:**

See above

---

> ### Author Response · Authors · 2025-11-25
> **Author Response (1/2)**
>
> > **W1: My main concern with the paper is the lack of advanced agent approaches evaluated on the benchmark. The baseline agent tested consists of a basic agent with a simple prompt, and in my understanding only represents the performance lowerbound on the benchmark. Without evaluating more advanced approaches, it is difficult to understand how well competitive agents would perform on the benchmark.**
>
> We thank the reviewer for pointing out the importance of evaluating more advanced agents. Our original submission focused on bare LLM agents to isolate the intrinsic difficulty of AgentSynth, but we agree that including stronger, scaffolded agents better contextualizes the benchmark.
>
> In the revised manuscript, we therefore added experiments with stronger, scaffolded agents built on the Agent S framework (**Section 5.4: Agent Scaffolding Evaluation**).
>
> Concretely, we evaluate [Agent S3](https://arxiv.org/pdf/2510.02250) (Gonzalez-Pumariega et al., 2025), which augments the backbone LLM with a plan-act-reflect loop. The agent first drafts a high-level plan, executes actions step-by-step while maintaining a scratchpad of state summaries, and periodically invokes a self-verifier to check whether subgoals have been achieved and to revise the plan when it detects failures. This scaffold is representative of competitive modern agent designs that combine reasoning, decomposition, and self-reflection. We run Agent S3 on multiple backbones (o4-mini, GPT-4.1, GPT-5, GPT-5.1) across all six difficulty levels.
>
> As shown in Table 1 below, Agent S3 substantially improves performance over the bare agents, especially on easier levels. For example, o4-mini improves from 18\% to 44\% success on level 1 and from 4\% to 20\% on level 6, GPT-4.1 improves from 14\% to 36\% on level 1 and from 0\% to 10\% on level 6. Stronger backbones (GPT-5 / GPT-5.1 with S3) achieve higher success rates (e.g., 62\% on level 1), but all configurations still exhibit a pronounced drop as difficulty increases, with success on level 6 about 20\%.
>
> **Table 1 - Success rate (\%) vs. difficulty level with Agent S3 scaffolding.**
>
> | Model                  | L1  | L2  | L3  | L4  | L5  | L6  |
> |------------------------|-----|-----|-----|-----|-----|-----|
> | o4-mini (bare)         | 18  | 14  | 10  |  8  |  4  |  4  |
> | gpt-4.1 (bare)         | 14  |  2  |  4  |  0  |  0  |  0  |
> | o4-mini + S3     | 44  | 34  | 26  | 26  | 18  | 20  |
> | gpt-4.1 + S3     | 36  | 18  | 14  | 14  | 10  | 10  |
> | gpt-5   + S3     | 42  | 28  | 22  | 20  | 20  | 18  |
> | gpt-5.1 + S3     | 62  | 32  | 20  | 22  | 16  | 14  |
>
>
> We additionally report results with [Agent S2.5](https://arxiv.org/pdf/2504.00906) (Agashe et al., 2025), an intermediate scaffold that includes planning and limited self-correction but fewer verification loops than S3 (Table 2). Agent S2.5 shows similar trends: consistent gains over bare agents, but persistent difficulty on higher levels, indicating that our findings are robust to the choice of scaffold.
>
> **Table 2 - Success rate (\%) vs. difficulty level with Agent S2.5 scaffolding.**
>
> | Model                   | L1  | L2  | L3  | L4  | L5  | L6  |
> |-------------------------|-----|-----|-----|-----|-----|-----|
> | o4-mini (bare)          | 18  | 14  | 10  |  8  |  4  |  4  |
> | gpt-4.1 (bare)          | 14  |  2  |  4  |  0  |  0  |  0  |
> | o4-mini + S2.5    | 42  | 34  | 30  | 24  | 26  | 20  |
> | gpt-4.1 + S2.5    | 40  | 24  | 14  | 16  | 12  | 10  |
> | gpt-5   + S2.5    | 46  | 28  | 28  | 18  | 22  | 20  |
> | gpt-5.1 + S2.5    | 60  | 36  | 26  | 20  | 18  | 16  |
>
> Overall, these new experiments demonstrate that AgentSynth remains challenging even for modern and advanced agent pipelines.
>
>
> > **W2: A missing citation is "Exposing Limitations of Language Model Agents in Sequential-Task Compositions on the Web. Furuta et al., TMLR 2024"**
>
> We thank the reviewer for pointing out this omission. In the revised manuscript, we have added this citation in the Introduction section.

---

> ### Author Response · Authors · 2025-11-25
> **Author Response (2/2)**
>
> With the evaluations using advanced agents, we believe we have addressed your central concern about the realism of the evaluation. Beyond the changes that directly respond to your comments, we also made several broader improvements that strengthen the manuscript as a whole:
>
> - **Asymmetry ablation (Sec. 5.5)**. We compared our information-asymmetric AgentSynth pipeline with a direct-instruction baseline that generates long tasks in one shot, confirming that asymmetry makes trajectory collection easier while producing significantly harder evaluation tasks.
>
> - **Base-generator ablation (Sec. 5.6)**. We replaced the base model for data generation with several different models. Results show that synthesized data remain stable across base models.
>
> - **Verifier calibration (Sec. 4.2)**. We calibrated the LLM verifier against human judgments and showed that its binary decisions and completion scores are well aligned with human labels, and that it is robust to subtle perturbations.
>
> - **Multi-axis task complexity analysis (Sec. 4.3)**. We analyzed how horizon, number of applications, context switches, memory span, and fine-grained perception all scale with difficulty level, showing that AgentSynth’s levels align with several complementary difficulty axes.
>
>
> We hope these additions clarify the strength and realism of the evaluation setup and demonstrate that the revised paper offers a richer and more robust study than the original submission.

---

### Meta-Review · Area_Chair_ZzCL · 2026-01-08

**Summary:**

**Summary.** This paper aims to automatically synthesize long-horizon, diverse tasks and trajectories. It introduces a pipeline named AgentSynth, leveraging six agents to step-by-step create challenging tasks based on subtasks.

**Strength (by reviewers).** important problem; clever, intuitive, and cost effective approach; practical & scalable; well-written paper; results supporting the usefulness of the synthesized data/benchmak; detailed statistical analysis

**Weakness(by reviewers).** lack of evaluation using advanced agents (e.g., agent scaffolding); limited causal evidence & ablation; narrow difficulty control; potential agent verification error; organization (ideally, more results but less analysis); "easy-to-hard" data synthesis is not entirely novel

**Decision.** The paper received an original rating of 4.67 (2, 6, 6). The authors provided an extensive rebuttal and revision; the rebuttal is easy to follow. Two reviewers responded to the rebuttal. Specifically, the score 2 reviewer raised the score to 4 but still expressed concerns about the novelty. The AC read the paper, reviews, and rebuttals. The AC found that a great portion of the concerns/questions have been addressed. Regarding novelty, while the "easy-to-hard" concept has been proposed, its realization for specific problem domains is not trivial, and this paper demonstrates a systematic way to realize it, which can likely be followed and further extended by future work. Overall, the paper has clear strengths and potential impacts; the AC also finds that its detailed analyses and empirical studies are quite valuable. The AC thus suggests acceptance.

**AC's further suggestions.**
- Include new results in the main paper
- Include the discussions that have not been incorporated into the revised paper into the camera-ready version, such as the answer to Q2 of Reviewer jFe8.
- Have a new subsection discussing novelty in the camera-ready version.

**Reviewer Concerns:**

**Reviewer kK3T.**
- **Addressed:** advanced agent approaches evaluated on the benchmark
- **Remained:** None

**Reviewer jFe8.**
- **Addressed:** limited causal evidence & ablation (to a great extent); narrow difficulty control (to a great extent); potential agent verification error (to a great extent); other questions
- **Remained:** None

**Reviewer CsTg.**
- **Addressed:** paper organization; lacks the use of any agent scaffold; further questions (to a great extent)
- **Remained:** novelty concern (partially addressed)

**Reviewer Scores:**

**Reviewer kK3T (6 to 7).** All the reviewers' questions/concerns have been addressed. The reviewer may keep the original score (6) or move it up to 8.

**Reviewer jFe8 (6 to 6).** The reviewer acknowledged that the rebuttal gives a clearer and more comprehensive understanding, and will keep the original score.

**Reviewer CsTg (2 to 4).** The reviewer acknowledged that the reviewer has no further questions, and would raise the score to 4. However,  concerns about the novelty prevent the reviewer from increasing the score further.

---

### Decision · Program_Chairs · 2026-01-26

Accept (Poster)